



# Evaluation of Total Column Water Vapour Products from Satellite Observations and Reanalyses within the GEWEX Water Vapor Assessment

Tim Trent[1], Marc Schröder[2], Shu-Peng Ho[3], Steffen Beirle[4], Ralf Bennartz[5], Eva Borbas[6], Christian Borger[7,4], Helene Brogniez[8], Xavier Calbet[9], Elisa Castelli[10], Gilbert P. Compo[11,12], Wesley Ebisuzaki[13], Ulrike Falk[7], Frank Fell[14], John Forsythe[15], Hans Hersbach[16], Misako Kachi[17], Shinya Kobayashi[18], Robert E. Kursinski[19], Diego Loyola[20], Zhengzao Luo[21], Johannes K. Nielsen[22], Enzo Papandrea[23], Laurence Picon[24], Rene Preusker[25], Anthony Reale[3], Lei Shi[26], Laura Slivinski[27], Joao Teixeira[28], Tom Vonder Haar[29], and Thomas Wagner[4]

[1]Earth Observation Science/National Centre for Earth Observation, School of Physics and Astronomy, University of Leicester, Leicester, UK
[2]Satellite-Based Climate Monitoring, Deutscher Wetterdienst, Offenbach, Germany
[3]NOAA/NESDIS/STAR, 5830 University Research Court, College Park, MD 20740-3818
[4]Satellite Remote Sensing Group, Max Planck Institute for Chemistry, Mainz, Germany
[5]Vanderbilt University, Nashville, TN, USA
[6]Space Science and Engineering Center, University of Wisconsin-Madison, WI, USA
[7]ECMWF, Robert-Schuman-Platz 3, 53175 Bonn
[8]Université Paris-Saclay, UVSQ, CNRS, LATMOS/IPSL, 78280, Guyancourt, France
[9]AEMET, C/ Leonardo Prieto Castro 8, Ciudad Universitaria, 28071 Madrid, Spain
[10]CNR-ISAC, Via Gobetti 101, 40129 Bologna, Italy
[11]University of Colorado, CIRES, Climate Diagnostics Center, Boulder, CO, USA
[12]NOAA Earth System Research Laboratory, Physical Sciences Division, Boulder, CO, USA
[13]NOAA/NCEP/CPC, 5830 University Research Court, College Park, MD 20740-3818
[14]Informus GmbH, 13187 Berlin, Germany
[15]Cooperative Institute for Research in the Atmosphere (CIRA), Colorado State University, Fort Collins, CO 80523
[16]European Centre for Medium-Range Weather Forecasts, Reading, UK
[17]Japan Aerospace Exploration Agency (JAXA), Tsukuba, Japan
[18]Numerical Prediction Development Center, Japan Meteorological Agency, Tsukuba, Japan
[19]PlanetiQ, 15000 West 6th Avenue, Suite 202, Golden, CO 80041
[20]Remote Sensing Technology Institute, German Aerospace Center, Oberpfaffenhofen, Germany
[21]Department of Earth and Atmospheric Sciences, City University of New York, City College, New York NY 10031
[22]Danish Meteorological Institute, Lyngbyvej 100, 2100, Copenhagen, Denmark
[23]CNR-ISAC, Via Gobetti 101, 40129 Bologna, Italy
[24]LMD/ IPSL,Sorbonne-université, CNRS, Paris, France
[25]Freie Universität Berlin, Kaiserswerther Str. 16-18, 14195 Berlin
[26]NOAA/NESDIS/NCEI, Asheville, NC, USA
[27]NOAA/OAR/PSL, 325 Broadway, Boulder CO 80305
[28]Jet Propulsion Laboratory, California Institute of Technology, Pasadena, California
[29]Department of Atmospheric Science, Colorado State University Fort Collins Colorado, USA

**Correspondence:** Tim Trent (t.trent@leicester.ac.uk)



**Abstract.** Since 2011 the Global Energy and Water cycle Exchanges (GEWEX) Water Vapor Assessment (G-VAP) has provided performance analyses for state-of-the-art reanalysis and satellite water vapour products to the GEWEX Data and Analysis Panel (GDAP) and the user community in general. A significant component of the work undertaken by G-VAP is to characterise the quality and uncertainty of these water vapour records to; i) ensure full exploitation and ii) avoid incorrect use or

interpretation of results. This study presents results from the second phase of G-VAP, where we have extended and expanded our analysis of Total Column Water Vapour (TCWV) from phase 1, in conjunction with updating the G-VAP archive. For version 2 of the archive, we consider 28 freely available and mature satellite and reanalysis data products, remapped to a regular longitude-latitude grid of $2° \times 2°$, and on monthly time steps between January 1979 and December 2019. We first analysed all records for a 'common' short period of five years (2005-2009), focusing on variability (spatial & seasonal) and deviation from

the ensemble mean. We observed that clear-sky daytime-only satellite products were generally drier than the ensemble mean, and seasonal variability/disparity in several regions up to 12 kg/m$^2$ related to original spatial resolution and temporal sampling. For 11 of the 28 data records, further analysis was undertaken between 1988-2014. Within this 'long period', key results show i) trends between -1.18±0.68 to 3.82±3.94 kg/m$^2$/decade and -0.39±0.27 to 1.24±0.85 kg/m$^2$/decade were found over ice-free global oceans and land surfaces respectively, and ii) regression coefficients of TWCV against surface temperatures

of 6.17±0.24 to 27.02±0.51 %/K over oceans (using sea surface temperature) and 3.00±0.17 to 7.77±0.16 %/K over land (using surface air temperature). It is important to note that trends estimated within G-VAP are used to identify issues in the data records rather than analyse climate change. Additionally, breakpoints have been identified and characterised for both land and ocean surfaces within this period. Finally, we present a spatial analysis of correlations to six climate indices within the 'long period', highlighting regional areas of significant positive and negative correlation and the level of agreement among records.

## 1  Introduction

Water vapour is the most important greenhouse gas in the Earth's climate system, acting as the predominant source of infrared opacity for the clear-sky atmosphere and (Douville et al., 2021). While directly and indirectly influencing radiative balance, surface fluxes and soil moisture, it is sufficiently abundant and short-lived that it is considered under natural control (Sherwood

et al., 2010). With prevalent positive feedback on the Earth's climate system ($\approx$2 W/m$^2$/K, Dessler et al. (2008)), water vapour exerts the largest amplification mechanism for anthropogenic climate change compared to radiative forcing from greenhouse gases (Held et al., 2000; Chung et al., 2014). Therefore, water vapour is a key parameter for the Earth's energy budget and climate analysis.

In 2011 the Global Energy and Water cycle Exchanges (GEWEX) water vapor assessment (G-VAP) was initiated by the

GEWEX Data and Analysis Panel (GDAP) with a remit to characterise the performance of state-of-the-art water vapour products to support these type of analyses. Therefore, the scope of G-VAP activities is to highlight the strengths, differences and





limitations of water vapour climate data records through consistent evaluation and intercomparison studies. The stability of long-term datasets is a key focus of the assessment. Through these activities, G-VAP supports the selection process of suitable water vapour data records by GDAP and the general climate analysis community (further details are available from

35 www.gewex-vap.org). Phase 1 of G-VAP concluded in 2017 with the publication of a World Climate Research Programme (WCRP) report (Schröder et al., 2017) and an archive of total column water vapour (TCWV), specific humidity and temperature profiles used within the analysis (Schröder et al., 2018).

In 2018, the assessment entered its second phase, focusing on the following objectives:

- The characterisation of water vapour data records,

- Informing users of issues within water vapour data products,

- Climate and process-oriented analysis,

- Continued link to key scientific questions and add more focus on process evaluation studies (PROES),

- Enhance regional analysis.

As in phase 1 of G-VAP, the assessment effort focuses on characterising the fitness of the various data records for climate

analysis. In this study, we present the evaluation of satellite and reanalysis of TCWV records collected for the second release of the G-VAP data archive. Section 2 briefly introduces the water vapour records that make the new archive. The methods used for evaluating the archive of TCWV records are outlined in section 3, with the results shown in section 4. Finally, we discuss our findings within the context of the assessment (section 5) and present our conclusions with recommendations in section 6.

## 2 Overview of version 2 of the G-VAP data archive

To support efforts within the second phase of the assessment, the G-VAP data archive has been updated to include new versions and products. It has been extended to cover the period from January 1979 to December 2019. 1979 was chosen as a starting point as it coincides with the launch of the NOAA-6 satellite, which carried a multispectral atmospheric sounding package which continues to the present day with various instrument modifications. Monthly mean fields for individual products have been processed onto a common spatial grid of 2°x2° and for where the archive temporal range exceeds original coverage values

are left undefined. The inclusion of a new flag within the archive files allows for valid data to be selected by the user. Further detail of the archive preparation is given in section 3.1. Within the time range covered by the archive, we define two main analysis periods:

- The five-year 'common period' between 2005-2009, which captures all products.

- The 'long period' runs from 1988-2014 and is designed to represent the Atmospheric Model Intercomparison Project

(AMIP) period from the latest sixth Assessment Report (AR6).



In the rest of this section, we detail which records are newly added products and which have been updated/extended. An overview of included products within version 2 of the G-VAP archive and analysis periods is shown in Figure 1. The archive is expected to be released with the $2^{nd}$ G-VAP report in 2023.

### 2.1 Updated and new records within version 2

The starting point for this study was the previous version of the G-VAP data archive (Schröder et al., 2018) with 22 satellite and reanalysis TCWV records. These records were initially split into three categories: i) records with no version update but extended time series, ii) records with a newer version (superseded), or iii) records with no updates in version or temporal coverage. From the initial set of records, 18 data sets fell into the first two categories, with 14 being retained and extended and four being updated to newer versions. In the case of the Moderate-resolution Imaging Spectroradiometer (MODIS) MYD08_M3 product,

we have also included the Terra version of the product (MOD08_M3). Short abstracts for these data sets are given in Schröder et al. (2018) and are not repeated here; however, details of these data sets are given in Table 1. Finally, ten further satellite and reanalysis records are added to the archive, with some including different versions of the same product if they were both available at the time of this study (e.g. AIRS v6 & v7). The following sections 2.1.1 to 2.1.8 provide abstracts for each new product in the archive.

### 75 2.1.1 AIRS

On board the National Aeronautics and Space Administration's (NASA) Aqua platform, the Atmospheric Infrared Sounder (AIRS) produces a range of geophysical products, including cloud-cleared radiances, temperature and water vapour profiles, cloud properties, methane, carbon monoxide, ozone, and surface temperature. While retrievals can be done with AIRS infrared (IR) radiances alone, this study utilises output from combined AIRS IR and microwave (MW) observations from the Advanced

Microwave Sounding Unit (AMSU). This configuration, known as the 'golf ball', collocates nine AIRS footprints (3x3) within an AMSU field of regard (FOR) and assumes the scene to be homogeneous except for cloud amount. The algorithm compensates for the cloud effects on the IR radiances within the scene before being passed to the final retrieval stages. Further details can be found in Susskind et al. (2003) and Manning et al. (2020). In this study, we use the AIRX3STM monthly mean TCWV record (AIRS Science Team/Joao Teixeira , 2013; AIRS project , 2019), which integrates the AIRS L2 water vapour profile

to create a column measurement before being averaged over the month on a 1°x1° grid for both the ascending (south to north pole) and descending (north to south pole) nodes of the orbit (day & night respectively). We have averaged both ascending and descending monthly data for the monthly mean used in this study. While the Aqua platform has been in orbit for 20 years, this data record only runs between the $30^{th}$ of August 2002 through to the $24^{th}$ of September 2016 due to the failure of the AMSU-A2 module.



### 2.1.2 CM SAF/WV_cci

The global TCWV data record combines microwave (MW) and near-infrared (NIR) imager-based TCWV data over the ice-free ocean and over land, coastal ocean and sea ice, respectively. The data record relies on MW observations from the Special Sensor Microwave Imager and Sounder (SSM/I, SSMIS), Advanced Microwave Scanning Radiometer for EOS (AMSR-E) and Tropical Rainfall Measuring Mission's Microwave Imager (TMI). Level 1b (L1b) MW used over global ice-free oceans

is partly based on a fundamental climate data record (Fennig et al., 2020) generated within the European Organisation for the Exploitation of Meteorological Satellites' (EUMETSAT) Satellite Application Facility on Climate Monitoring (CM SAF). While over land surfaces, NIR L1b measurements from Modern Era Retrospective-Analysis for Research and Applications (MERIS, 3rd reprocessing), MODIS-Terra (collection 6.1) and Ocean and Land Colour Instrument (OLCI, 1st reprocessing), are used. The NIR and MW TCWV observations are combined within the Water Vapour project of the European Space

Agency's (ESA) Climate Change Initiative (WV_cci). Details of the retrievals are described in Andersson et al. (2010) and Graw et al. (2017) for the MW imagers as well as in Lindstrot et al. (2012), Diedrich et al. (2015) and Fischer et al. (2021) for the NIR imagers. The atmosphere's water vapour is vertically integrated over the full column and given in units of kg/m$^2$. The MW and NIR data streams are processed independently and combined afterwards by not changing the individual TCWV values and their uncertainties. The data record has a spatial resolution of 0.5°x0.5° and 0.05°x0.05°, with the NIR-based

data being averaged and the microwave-based data being oversampled to match the lower, respectively higher spatial resolution. The product is available as daily and monthly means and covers the period from July 2002 – December 2017. The data is subsequently referred to as the CM SAF/WV_cci data record and is referenced and accessible via the following DOI: https://doi.org/10.5676/EUM_SAF_CM/COMBI/V001.

### 2.1.3 GOME EVOL

The 'GOME evolution climate' product was generated within the GOME-Evolution project funded by ESA, and the retrieval is described in detail in Beirle et al. (2018). It is based on measurements from the satellite instruments Global Ozone Monitoring Experiment (GOME), Scanning Imaging Absorption Spectrometer for Atmospheric Chartography (SCIAMACHY), and GOME-2 in the red spectral range, using the retrieval proposed in Wagner et al. (2003, 2006), with all satellites measurements occurring around 10:00 hours local time. As stated in Beirle et al. (2018), a particular focus of the climate product is the

consistency amongst the different sensors to avoid jumps from one instrument to another. This is reached by applying robust and simple retrieval settings consistently. Potentially systematic effects due to differences in ground pixel size are avoided by merging SCIAMACHY and GOME-2 observations to GOME spatial resolution, allowing for a consistent treatment of cloud effects. In addition, the GOME-2 swath is reduced to that of GOME and SCIAMACHY to have consistent viewing geometries. The remaining systematic differences between the sensors are investigated during overlap periods and corrected in the homog-

enized time series. The 'GOME evolution climate' product contains monthly mean TCWV from July 1995 to December 2015 on 1° spatial resolution. It is available at https://doi.org/10.1594/WDCC/GOME-EVL_water_vapor_clim_v2.2.



### 2.1.4 ERA5

The European Centre for Medium-Range Weather Forecasts (ECMWF) Reanalysis v5 (ERA5) atmospheric general circulation model and 4D-Var assimilation system are based on the IFS Cycle 41r2 and IFS Cycle 41r2 4D-Var versions of the ECMWF integrated forecast system (IFS). It is conducted at a resolution of about 31 km in the horizontal and 137 levels in the vertical from the surface to 0.01 hPa, and the analysis is available at a 1-hour temporal resolution. ERA5 is the successor of the ERA-Interim reanalysis. Among others, ERA5 exhibits significant improvements in resolution, including a 10-member ensemble of data assimilation and various additional parameters, and incorporates an improved 4D-Var and variational bias correction scheme through the utilisation of (newly reprocessed) datasets and recent instruments. ERA5 exploits a vast amount of datasets, including *in situ* measurements from land stations as well as measurements from ships and drifting buoys, radiosondes, pilot balloons, aircraft, and wind profilers. The largest amount of data comes from satellite observations, focusing on polar-orbiting and geostationary satellites and improvements in all-sky assimilation. ERA5 provides complete atmospheric products globally from 1950 onwards at a mixed hourly/3-hourly output frequency and is continued with updates in near real-time. ERA5 and its quality are described in Hersbach et al. (2020). Monthly means of TCWV with a spatial resolution of 2°x2° were downloaded from https://cds.climate.copernicus.eu/#!/home in November 2020.

### 2.1.5 MODIS TIR NASA (Aqua & Terra)

In addition to the MODIS NIR TCWV data, the second TCWV record based on NASA's Moderate Resolution Imaging Spectroradiometer (MODIS) used in this study is the atmospheric profile product MYD07_L2/MOD07_L2 (Aqua/Terra respectively). This product uses the Thermal Infrared (TIR) bands 25 and 27 through 36 to retrieve temperature and moisture profiles, total-ozone burden, atmospheric stability, and atmospheric water vapour. The level 2 (L2) product contains the geophysical parameters at a resolution of 5x5 km for both clear-sky day & night scenes. A scene is considered clear if at least nine 1x1 km pixels are cloud-free, for which the MODIS cloud mask (MOD35_L2) is used for screening. The retrieval algorithm uses a modified version of the International Television and Infrared Observation Satellite (TIROS) Operational Vertical Sounder (TOVS) Processing Package (ITPP) for which the initial state vector uses a linear regression first-guess approach (Seemann et al., 2003, 2006). Monthly mean fields gridded at 1°x1° are available within collection 6.1 of the MOD08_M3/MYD08_M3 L3 product, which is also used for the source of the MODIS NIR TCWV product from version 1 of the archive. The level-3 MODIS MYD08 and MOD08 products can be obtained from the NASA Level-1 and Atmosphere Archive and Distribution System (LAADS) Distributed Active Archive Center (DAAC), Goddard Space Flight Center, Greenbelt, MD, USA (Platnick et al. (2015), https://doi.org/10.5067/MODIS/MYD08_D3.061, http://dx.doi.org/10.5067/MODIS/MOD08_M3.061)

### 2.1.6 MPIC OMI

The TCWV data set provided by the Max Planck Institute for Chemistry (MPIC) is based on hyperspectral satellite measurements in the visible blue spectral range from the Ozone Monitoring Instrument (OMI) onboard NASA's Aura satellite with an equator crossing time of about 13:30 local time. For this data set, the retrieval algorithm of Borger et al. (2020) has been



modified to account for the specific instrumental issues of OMI (e.g. the "row-anomaly") and the inferior quality of solar
reference spectra. The TCWV data set only includes measurements for which the effective cloud fraction < 20%, the AMF
(air mass factor) > 0.1, the ground pixel is snow- and ice-free, and the OMI row is not affected by the "row-anomaly" over
the complete time range of the data set. The remaining satellite measurements are then binned to a regular latitude-longitude
lattice via an area-weighted gridding algorithm, covering both land and ocean surfaces globally. In-depth details about the data
set generation, sampling errors, and clear-sky bias are available in Borger et al. (2023a). Moreover, the data set has been val-
idated with respect to reanalysis data, satellite measurements, and radiosonde observations. Furthermore, a temporal stability
analysis demonstrated that the MPIC OMI TCWV data set shows no significant deviation trends and aligns with the GCOS
stability requirements. The water vapour of the atmosphere is vertically integrated over the full column and given in units of
kg/m$^2$. The data set has a spatial resolution of $1° \times 1°$ and is available as monthly means for the time range January 2005 to
December 2020. The data set is referred to as the MPIC OMI TCWV climate data record and is available via the following
DOI: https://doi.org/10.5281/zenodo.5776718 (Borger et al., 2023b).

### 2.1.7   NCEP-DOE 2

The National Center for Environmental Prediction (NCEP), Department of Energy (DOE) Reanalysis 2 (NCEP-DOE 2) was
originally created to support the $2^{nd}$ Atmospheric Model Intercomparison Project (AMIP). The NCEP-DOE 2 uses a global
spectral model with a spatial resolution of T62 ($\approx$ 210 km) and 28 vertical levels for both forecasts and analyses. The data
assimilation uses spectral statistical interpolation (or 3D-Var) with a one-way coupled ocean model 4D assimilation (Kalnay
et al., 1996). Outputs are available daily on 6 hourly time steps, or as daily or monthly averages. In addition to the spectral
T62 grid, geophysical parameters are also available on a regular $2.5°$ latitude x $2.5°$ longitude global grid. NCEP-DOE 2 has
a number of improvements over its predecessor, with corrections for the southern Hemisphere bogus data (PAOBS) problem
between 1979-1992, snow cover and snow melt, humidity diffusion and discontinuities, and oceanic albedo. Further details on
the improvements and updates made to NCEP-DOE 2 can be found in Kanamitsu et al. (2002). One important point regarding
NCEP-DOE 2 is that it does not assimilate either SSM/I, SSMIS data or TOVS/ATOVS water vapour profiles to constrain
atmospheric moisture. NCEP-DOE 2 data are provided by NOAA PSL, and was last accessed via https://psl.noaa.gov in May
2021.

### 2.1.8   NOAA 20CR V2C and V3

The NOAA-CIRES-DOE (National Oceanic and Atmospheric Administration, Cooperative Institute for Research In Envi-
ronmental Sciences) Twentieth Century Reanalysis (20CR) project provides a comprehensive dataset of reconstructed global
weather spanning over 150 years at sub-daily resolution. The analysis is generated by assimilating only surface pressure
observations into the NCEP Global Forecast System (GFS) Model with prescribed sea surface temperatures and sea ice con-
centrations. Currently, two versions of 20CR are available: i) version 2c (V2c), which runs from 1851 to 2014 on 6 hourly
time steps on a $2°$x$2°$ resolution grid, and ii) version 3 (V3) that has an extended coverage spanning from 1836 to 2015, and
a high temporal and spatial grid of 3 hours and $1°$x$1°$ respectively (Slivinski et al., 2019, 2021). Both product versions use a



'deterministic' Ensemble Kalman Filter (EKF) for the data assimilation (Whitaker et al., 2004; Compo et al., 2011); however, V3 also includes an additional 4-dimensional incremental analysis update (Lei & Whitaker 2016). NOAA PSL provided the NOAA 20CR data and was last accessed via https://psl.noaa.gov in November 2020.

## 3 Methods

This section describes the preparation and analysis methodologies used in this study.

### 3.1 Preparation of TCWV records for inclusion in the archive

For each data record used in this study, monthly mean TCWV fields were first processed onto a common grid format of $2°$x$2°$ between January 1979 and December 2019. The data sets were downloaded at their native resolutions and processed in one of two ways. For the reanalysis records, each monthly gridded TCWV global field was first shifted in longitude space to run between $-180°$ to $180°$ before being linearly interpolated onto the centres of the archive common grid. The monthly mean Level 3 (L3) satellite products were regridded if the native resolution was less than that of the common grid and averaged if separated into day/night means. All data were then written in netCDF format, with a new time flag added to indicate monthly time steps where valid data exists.

### 3.2 Calculation of trends

As with the first phase of G-VAP (Schröder et al., 2019), a main metric used to characterise the performance of records in the archive is the TCWV trend over the 27-year 'long period'. It is important to note that trends estimated within G-VAP are used to identify issues in the data records rather than analysis of climate change. Before trends are calculated, global TCWV time series of ice-free oceans and land surfaces are created. For this step, a conservative mask (Figure 2) is applied to select either ice-free ocean or land grid cells between $±60°$ at each monthly time step. The weighted mean TCWV at time step $t$ ($\overline{TCWV}_t$) is then calculated from all valid data points ($P$):

$$\overline{TCWV}_t = \frac{\sum_{i=1}^{P} TCWV_{t,i} \times w_i}{\sum_{i=1}^{P} w_i}, \qquad t = 1, 2, 3, \ldots, N, \tag{1}$$

where the weights $w$ are the cosine of the latitude. Next, a level shift linear regression model (Weatherhead et al., 1998) is used to calculate the trend:

$$\overline{TCWV}_t = \mu + \omega X_t + \delta U_t + \eta_t, \qquad t = 1, 2, 3, \ldots, N, \tag{2}$$

where $\mu$ is the intercept, $\omega$ is the trend, $X_t$ is the time index, $\delta$ is the magnitude of any shift, $U_t$ is the step function, and $\eta_t$ is the fit residual. For the purposes of performance analysis, the step function is assumed to be zero. However, break points in the TCWV times are characterised later on. For the residuals, the same approach is used from Schröder et al. (2019) where the El Niño Southern Oscillation (ENSO) strength and annual cycle are fitted simultaneously. The Japan Meteorological Agency (JMA) index (Bove et al., 1998), which is calculated from sea surface temperature (SST) anomalies, is used as a consistent





source for the ENSO strength across all records. Figure 3a illustrates the median contribution of ENSO to TCWV variability across the thirteen records spanning the long period. Positive values represent regions which see positive/negative TCWV changes for the El Niño and La Niña phases of ENSO, respectively, while the negative regions experience opposite behaviour with increases in TCWV during La Niña and a decrease during El Niño. The spread in the ENSO contribution is shown in
Figure 3b, represented by the median absolute deviation (MAD):

$$MAD = median(E_{\lambda,\phi} - \overline{E_{\lambda,\phi}}), \tag{3}$$

where $E$ is the ENSO weight strength as a function of longitude ($\lambda$), and latitude ($\phi$), and $\overline{E}$ is the median ENSO strength. The largest variability is seen in the tropics, with ENSO regions seeing 10-20% variability between data records.

In addition to the calculation of trends, a regression against surface temperature is performed on each data record following
the approaches outlined in Dessler and Davis (2010) and Mears et al. (2007). Over oceans, sea surface temperature (SST) data from the European Space Agency (ESA) Climate Change Initiative (CCI) is used (Merchant et al., 2019; Merchant and Embury, 2020). Surface air temperature (T2m) from the European Centre for Medium-Range Weather Forecasts (ECMWF) ERA5 reanalysis (Hersbach et al., 2020) is used for both ocean and land surfaces.

### 3.3 Detection of breakpoints

The detection of breakpoints is carried out as described in Schröder et al. (2019). Here, a summary of the approaches is provided, and a few newly implemented changes are also mentioned. Two breakpoint analyses are applied: the Penalised Maximal F (PMF) test (Wang, 2008a, b) and a variant of the standard normal homogeneity (SNH) test (Hawkins, 1977; Alexandersson, 1986), as proposed in Reeves et al. (2007). The breakpoint analyses detect abrupt changes in the time series of TCWV, and the output of this analysis is the time and the strength associated with the breakpoint. Here, breakpoints are
provided when the level of significance is 0.05. Two tests are applied to increase confidence in case both tests observe the same breakpoint. Input to the PMF and SNH tests are anomaly differences, i.e., after removal of the mean annual cycles, the difference between the anomalies from a data record and a reference are computed. Over the ocean, HOAPS V4 is used as a reference; elsewhere, ERA5 was used. Using HOAPS V4 and ERA5 as references is not meant to be a sign of superior quality. Further details and comments are given in Schröder et al. (2019) and Schröder et al. (2016). The UW HIRS v2.52 data
record contains undefined values in the period October - December 1990. This gap has been linearly interpolated to allow the application of the homogeneity analysis. The interpolated data is not included in any other analysis nor in the time series plots.

### 3.4 Correlation to climate indices

The final test applied to archive records covering the 'long period' is to calculate their correlation to various climate indices. With TCWV records now spanning Multiple decades, provides new opportunities to go beyond trends and study global climate
signals and phenomena, e.g. teleconnections (Wagner et al., 2021). Therefore, it is vital to understand the representativeness and correlations of climate signals between different data records. The indices chosen for this study are:





– **Atlantic Meridional Mode (AMM)**: AMM is the dominant source of coupled ocean-atmosphere variability in the Atlantic.

– **Arctic Oscillation (AO)**; AO is the shifting atmospheric pressure back and forth between the Arctic and mid-latitudes areas of the North Atlantic and Pacific Oceans.

– **North Atlantic Oscillation (NAO)**: NAO is based on the surface sea-level pressure difference between the Subtropical (Azores) High and the Subpolar Low.

– **El Nĩno Southern Oscillation Index 3.4 (NINO3.4)**: NINO3.4 is one of several El Niño/Southern Oscillation (ENSO) indicators based on sea surface temperatures. NINO3. 4 is the average sea surface temperature anomaly in the region bounded by 5°N to 5°S, from 170°W to 120°W.

– **Pacific Decadal Oscillation (PDO)**: PDO is characterised by a change in sea surface temperature in the North Pacific (north of 20°N). The change usually occurs abruptly. PDO has a larger frequency than ENSO.

– **Pacific Meridional Mode (PMM)**: PMM is defined as the leading mode of non-ENSO coupled ocean/atmosphere variability in the Pacific basin.

These indices are the same as those used in phase 1 of the G-VAP assessment with the addition of the PMM. Further details of each index are described in Table B1 and the time series plotted in Figure S1. For each record, the Pearson correlation ($r$) is calculated between the TCWV anomalies ($\Delta WV$) and the specific climate index ($CI$) for each common grid cell:

$$r = \frac{\sum_{t=1}^{N}(CI_t - \mu_{CI})(\Delta WV_t - \mu_{\Delta WV})}{\sqrt{\sum_{t=1}^{N}(CI_t - \mu_{CI})^2 \sum_{t=1}^{N}(\Delta WV_t - \mu_{\Delta WV})^2}} \tag{4}$$

where $CI_t$ and $\Delta WV_t$ are the climate index and TCWV anomaly at time step $t$, and $\mu_{CI}$ and $\mu_{\Delta WV}$ are the mean climate index and TCWV anomaly respectively. A spatial analysis is then performed between the different records to assess the level of agreement in correlation results for each climate index. Finally, it is important to note that correlations also exist between the different climate indices; this is illustrated in Figure 4.

## 4 Results

Results from the analysis of TCWV records are presented here and have been split into two distinct sections; the first covers the short 'common' period, and the second is a broader analysis of the longer-term records.

### 4.1 Evaluation of records within the common period

Figure 5 presents seasonal TCWV maps of the archive ensemble mean ($\mu$), standard deviation ($\sigma$), coefficient of variance ($CV = 100 * \sigma/\mu$), and range (TCWV$_{max}$-TCWV$_{min}$) for the whole common period (2005-2009) for all 28 records. The





seasons are defined as DJF (northern hemisphere (NH) winter), MAM (NH spring), JJA (NH summer), and SON (NH autumn)
where the capitalised letter of each month (e.g. December, January, February) makes up the season acronym.

The column of maps presents the seasonal ensemble means, where we can observe a clear seasonal cycle with no anomalous
regions immediately apparent. However, when we examine the standard deviation for each season ($2^{nd}$ column), four regions
stand out with high absolute values:

- The first is the Sahel, where we observe standard deviations $\geq 10$ kg/m$^2$ between MAM and SON.

- The second region is the Indo-Gangetic Plain (IGP), where JJA and SON values are well above 10 kg/m$^2$, exceeding 15
  kg/m$^2$ in some regions.

- The northeast of China and Japan is the third region, where $\sigma$ values are approximately 12 kg/m$^2$ for NH summer and
  autumn months.

- The fourth and final region is observed in the USA along the Californian coast. In this narrow region, variability amongst
the data records is between 11-12 kg/m$^2$ for JJA and SON.

Over the tropics and mid-latitudes, we see variability from cloud cover, which results in standard deviation values between
5-7 kg/m$^2$. In polar regions, these drop to $\sigma$ values of $\leq 1$ kg/m$^2$. However, we get a slightly different perspective on the TCWV
variability by normalising the standard deviation with the seasonal mean. The third column of Figure 5 shows CV, where now
three different areas are highlighted (where CV values >70%). These regions are i) the Tibetan plateau, ii) the Andes, and iii)
high latitudes/Polar regions. The common theme between these three is that all have dry atmospheres and variable topography,
especially for the Tibetan plateau and the Andes. The CV has a distinct seasonal signal for Polar regions, with the hemispherical
winter exhibiting the greatest variability among the data records.

Examining the seasonal range of the archive records (Figure 5, $4^{th}$ column ) reveals that areas with high standard deviations,
in general, have a range of $\geq 50$ kg/m$^2$, while regions with high CV see seasonal ranges of 10-20 kg/m$^2$. For very dry regions,
10-20 kg/m$^2$ can be 3-10 times the season mean TCWV, whilst for tropical regions, high standard deviations are associated
with high monthly mean TCWV values. Therefore, this highlights a significant disagreement between the archive records for
dry atmospheres, especially at high latitudes.

The variability of TCWV observed within the archive is investigated further by comparing each record to the ensemble
mean for the 'common period'. For each dataset, the monthly difference to the ensemble mean has been calculated before
being averaged over the time period. Next, these differences were summed to calculate each dataset's global value. Finally,
all results were sorted from driest to wettest and shown in Figure 6. It should be noted that here, the ranking of datasets is
not a statement of climate performance (i.e. bias to characterise reference/truth) but rather highlights characteristics related to
differences in the observational/assimilation systems. From these global mean differences, the initial inferences we can make
are:

305   - 11 datasets are drier than the ensemble mean, while the remaining 17 are wetter, with the EMiR record exhibiting the
  smallest global $\Delta$TCWV.



- – All IR and NIR TCWV records have the driest global ΔTCWV, while other MW, MW+IR, & MW+NIR records sit between -2.8% to 4.3%.

- – The GOME Evolution and MPIC OMI products, which use the visible (blue or red) spectral region to retrieve TCWV, show very different results. Both being daytime-only products (morning vs afternoon overpass respectively), GOME Evolution has a ΔTCWV of 0.74% while MPIC OMI is much larger at 9.2%.

- – Overall, reanalyses are wetter than the global ensemble mean TCWV except for JRA55 and ERA 20C.

We also analysed ΔTCWV as global distributions relative to the ensemble mean over the whole common period to expand upon these singular global values. Shown in Figure 7, global maps of ΔTCWV reveal that the Intertropical Convergence Zone (ITCZ), Polar, storm track, and low stratiform cloud (as defined in Klein and Hartmann (1993)) regions are all key regions in the observed differences. Relating to the singular global values, these results also show:

- – IR satellite records show general dryness relative to the ensemble mean, except over regions with low stratiform clouds where wetter differences are observed.

- – MODIS NIR TCWV, like the IR products, are drier over oceans. However, overland regions, especially South America and North Africa, are wetter than the ensemble mean. The Tropical Warm Pool is a key region, where ΔTCWV values can go as low as -14%.

- – The visible records show very different patterns, especially in the Tropics. While the GOME Evolution product shows a general dry difference to the ensemble mean TCWV in the Tropics to mid-latitudes, the MPIC OMI dataset is much wetter. These differences can be attributed to i) spectral range (GOME Evolution uses the strong absorption band in the red part of the spectrum while MPIC OMI uses the blue spectral range with much weaker absorption bands, ii) overpass times between GOME/SCIA and OMI are different (differing cloud amounts), and iii) MPIC OMI is less sensitive to surface albedo changes across different surfaces, while for GOME Evolution the surface albedo differs strongly between ocean and land.

- – ERA 20C and JRA55 reanalyses are drier between ±30° (especially land surfaces) compared to other reanalysis datasets. One exception to this is NCEP-DOE 2 reanalysis which has a drier ITCZ region over ocean surfaces.

## 4.2 Evaluation of records within the long period

For this second set of analyses, we now concentrate on 13 data records within the archive which cover the 'long' period (1988-2014). From this set of products, 9 are from reanalyses, and the final 4 are satellite records. This analysis focuses on trends (including regression against surface temperatures), breakpoints within the global ocean and land time series, and correlation to climate indices. For trends, the values reported by G-VAP are not a statement of climate change but rather an indicator of comparative performance relative to other records in the archive.





### 4.2.1 Analysis of trends

We begin by examining global ice-free ocean and land surface trends between $\pm 60°$. TCWV anomaly time series were calculated for both regions by applying the conservative sea ice mask (Figure **??**) and filtering data, respectively. Weighted means

were calculated for each monthly time step before the trend was calculated using equation 2. Trend values have been ordered from minimum to maximum based on ice-free ocean results and are shown in Figure 8. The $1\sigma$ uncertainty is also calculated and shown as an error bar for each trend. From the spread of results, we see that:

- Over oceans, trends range from -1.18±0.68 to 3.82±3.94 kg/m$^2$/decade (NNHIRS, UWHIRS respectively) and between -0.39±0.27 to 1.24±0.85 kg/m$^2$/decade over land (NNHIRS, UWHIRS).

- Excluding the HIRS records, trends ranges become 0.12±0.17 to 0.94±0.92 kg/m$^2$/decade (ERA-Interim, NCEP CFS-R/CFSv2), and 0.11±0.15 to 0.53±0.32 kg/m$^2$/decade (ERA5, NCEP CFSR/CFSv2) for ocean and land respectively.

In addition to these trends, Figure 8 also gives the regression of each TCWV record against SST and T2m surface temperatures. For ice-free ocean regression coefficients, we also include the expected theoretical range described in Wentz and Schabel (2000). As with the first phase of G-VAP (Schröder et al., 2019), we see large differences between the records within the

350 archive. The first noticeable result is that the HIRS records exhibit more extreme behaviours relative to the other datasets. For NNHIRS, regression against ERA5 T2m gives the values -13.46±1.19 & -1.96±0.43 %/K over ocean and land, respectively. Regression against ESA CCI SST over ocean shows a similar result to ERA5 with a value of -13.17±1.17 %/K. The UWHIRS product displays the largest regression coefficients with values of 92.81±1.02 & 88.28±1.04 %/K over ice-free ocean surfaces (ERA5 T2m, ESA CCI SST respectively) and 24.94±0.23 %/K over land (ERA5 T2m). For the other products, the main results

we observe are:

- For regression coefficients over ice-free oceans we found ranges of 6.77±0.24 to 27.02±0.51 %/K for ERA5 T2m, and 6.17±0.24 to 24.17±0.45 %/K against ESA CCI SST.

- Over land surfaces, the range of regression coefficients are between 3.00±0.17 to 7.77±0.16 %/K against ERA5.

- Accounting for the uncertainty on the regression coefficients against ERA5 T2m and ESA CCI SST, we find that ERA-

360 Interim, MERRA2, NCEP-DOE 2, and ERA5 fall within the theoretical range for both temperature datasets. When only taking into account ERA5 T2m, the JRA55 product also falls within this range.

Examination of the spatial distributions of TCWV trends is used to investigate the inter-variability of these global trends further. Figure 9a presents the median absolute deviation (MAD) of 'long' period trends between the respective archive datasets. From looking at all records in this way, we can see three key features:

- Sea-ice boundaries show up as noise at high latitudes.

- Tropical oceans exhibit high levels of variability.




– Regions of high variability are observed over South America and central Africa.

The first feature is easily accounted for by applying the sea-ice mask to the satellite records impacted by sea-ice. From Figure 9b, we can observe that the mask successfully removes the previous noise.

Figure A1 presents the spatial distribution of TCWV trends for the 13 long-term records. The outlier behaviour observed in IR HIRS records is clearly driven by strong positive/negative trends over low stratiform cloud regions between $\pm 40°$.

### 4.2.2 Analysis of breakpoints

A high level of stability is a key requirement in case long-term data records are utilised in climate change analysis. Different approaches can be applied to assess stability. Here, the analysis of breakpoints in kg/m$^2$ at a specific time is carried out. As a

first step, as with the trends, the TCWV anomaly time series are computed. Input to the breakpoint analysis is then the anomaly difference to HOAPS v4 over ocean and ERA5 over land (see section 3.3 for details).

Figures 10 and 11 show the anomaly time series, shifted by the data record's mean value and breakpoints observed over the ocean and land, respectively. The spread among the data records is approximately 3 kg/m$^2$, with the exception of the two HIRS-based data records. The breakpoint analysis identified 38 breakpoints over the ocean, 21 identified by both homogeneity

tests and 13 breakpoints over land, 9 observed by both tests. Only two out of nine break points observed over land are also present over the ocean when applying a match criterium of $\pm 3$ months. It is recalled that the homogeneity analysis is applied unsupervised and consistently to all data records such that the results for the different data records are comparable. The breakpoint analysis might not necessarily identify breakpoints correctly, in terms of their presence and strength, and might also miss breakpoints. An example of the latter is a seemingly undetected breakpoint in the NNHIRS record over land in late

1989. A possible reason is that breakpoints close to the start and stop times of the data records are difficult to detect. When looking at the anomaly time series, the NNHIRS data record exhibits a decrease in TCWV over ocean until approximately 1998 while TCWV from UWHIRS decreases over land after 2001, both features in contrast to the other data records. Both seem to be affected by stability issue, that is, by a change in bias over time. Noteworthy are also anomaly features in the time series of NNHIRS over land in the period 1990-1995.

The time and strength of detected breakpoints are given in Tables 2 and 3, together with a potential explanation for the presence of such breakpoints. It is noted that the given explanations are actually temporal coincident between observed breakpoints and changes in the observing system. However, a physical explanation has not been explored and is thus not provided here. In the majority of cases, the observed breakpoint coincides with a change in the observing system. In 20% of the cases with confirmed breakpoints, a potential reason for the breakpoint is unknown. The observed breakpoints are generally small, except

for the breakpoints observed in UWHIRS. Most breakpoints are observed in ERA5 over the ocean. It is emphasised that the anomaly difference time series between HOAPS v4 and ERA5 exhibits a very low noise level, and only then can the breakpoint analysis detect small breakpoints. However, only one breakpoint of ERA5 was confirmed with the second homogeneity test. In contrast, the strong breakpoints in UWHIRS reduce the ability of the applied methodology to detect additional small breaks, for example, in 2002 over the ocean (see figure 10). If a break is caused by the utilised references (HOAPS v4 over ocean and





ERA5 over land), such a break would need to be present in most data records. This seems not to be the case. The breakpoint over the ocean between REMSS V7 and HOAPS V4 in 2001-07 does not coincide with a change in the observing system. The break coincides with a small increase and anomaly present in most data records.

Finally, the largest and smallest mean trend estimates shown in Figure 8 are observed for NNHIRS and ERA-Interim (smallest, i.e., negative trends) and UWHIRS and NCEP CFSR/CFSv2. In these cases, the observed breakpoints are either predominantly strongly positive or negative and explain the unexpectedly large and small trend estimates.

### 4.2.3 Correlation to climate indices

For the final analysis, TCWV anomalies for each long-term record were compared to the seven climate indices detailed in Table B1. An example of NAO, NINO34 and PDO correlation maps is shown in Figure 12, with statically significant correlations indicated with hatching. Figures for each data record with all seven climate indices can be found in the supplementary material. Next, these correlations were inter-compared between the datasets for each climate index. Notably, we identified regions where at least 50% or 100% of the records agreed on either positive or negative correlations. The absolute values differ over land and ocean surfaces as the MW satellite records provide only data over ice-free global oceans. The results from this test are shown in Figure 13, with negatively correlated regions shown on the left-hand side in orange, while regions of positive correlation are presented on the right-hand side in purple. The key results from these comparisons show:

– Generally, positive correlations between all datasets occur in expected regions related to the specific index.

– We also observed positive correlations between all water vapour records off the coast of Antarctica related to ENSO3.4 and the southern ocean and tropical warm pool regions with AMM.

– Ambiguity in the agreement among data records is also found over Africa for PDO, the Pacific Ocean for NAO, and over the African and Antarctic continents for PMM.

– Consistent negative correlations with PDO within the archive are observed over North West America and Eastern Europe.

– Over Greenland and North America, all water vapour records agree on a negative correlation to NAO and AO.

– However, we observe an ambiguity in negative correlations above 80°N for PMM, the broader Pacific regions for NINO3.4 and PDO, and over Africa, South America, and ocean regions for tropical latitudes for AO and NAO.

## 5 Discussion

This study partly continues the G-VAP activities carried out within its first phase. Approaches used here differ from approaches used in the first phase of G-VAP. The archive contains more members, particularly from reanalysis and HIRS; different periods were considered for common period and long-term analysis and a new land/sea and sea-ice mask was used. Also, new versions of HOAPS and ERA served as references for breakpoint detection and different SST and $T_{2m}$ data records were employed for





regression analysis. All this might provide a reason for differences to results from G-VAP's first phase. Nonetheless, this study
generally confirms the results from the first G-VAP phase (Schröder et al., 2016, 2019):

- The data records exhibit distinct spatial features in terms of biases, standard deviation, and mean absolute trend differences. Again, South America and Central Africa stick out and additional regions: Sahel, IGP, and parts of China and Japan.

- Large differences in trend and regression estimates occur over the ocean.

- The majority of data records are affected by breakpoints.

- The occurrence of breakpoints seems to have a regional dependency (here, ocean versus land), and in most cases, the breakpoints coincide with changes in the observing system.

Subsetting the ensemble by removing IR-based products removes differences and variability significantly, in particular over tropical oceans. This is interpreted to be dominated by the instability of the underlying HIRS data records.

The number of data records that exhibit regression values within the expected range over ocean is larger among data records from the G-VAP data archive 2 than from the G-VAP data archive 1. This can likely be explained by the archive 2 containing mainly additional reanalysis data records. It is recalled that various assumptions apply for the estimation of the theoretical expectation and that violations of these assumptions can give reason to larger than expected regression values (Mieruch et al., 2014). Additionally, Shi (2018) observed an ocean-basin dependent time lag between SST and TCWV. Results shown in Wentz
and Schabel (2000) indicate that the lag can also be a function of event and with that time. Even more so, the local response to ENSO also exhibits variability, as observed by Stephens et al. (2018) for precipitation. The presence of time lags between SST and TCWV during El Niño events was not considered during computation of regression. Following discussions in Falk et al. (2022) the expectation over land is additionally affected by the potential limitation of water vapour fluxes into the atmosphere (Byrne, 2016). Such an input flux depends on various processes and parameters, among others, advection from ocean to land,
presence of surface water, soil moisture, and other factors. Byrne (2016) conclude that the moisture transport from the ocean is the dominating process for changes in specific humidity over land and evapotranspiration processes cause changes in relative humidity over land. The presence of increased SST and TCWV over the ocean during El Niño events might lead to a larger transport of moisture from ocean to land. At the same time, increased surface temperature might not be present over land and can thus lead to a reduced correlation between land surface temperature and TCWV.

Looking at the differences between individual data records to the ensemble mean reveals that the data records that rely on retrievals predominantly applicable to clear-sky conditions exhibit biases to the ensemble mean over the ITCZ and storm tracks, i.e., the bias coincides with predominantly cloudy regions.

The retrievals of TCWV are based on measurements obtained in different parts of the electromagnetic spectrum: among others, the visible, the near-infrared (NIR), infrared (IR), and microwave frequencies. Except for microwave observations, all
460 related retrievals are predominantly applied under clear-sky conditions. Though instantaneous water vapour products can show high quality and low uncertainty, this is not necessarily true for the gridded and temporally averaged products: Conditions



in clouds are typically more humid than the surrounding clear-sky areas (e.g., Fetzer et al. (2006)), and are not taken into account by the retrieval's clear-sky observations. This causes a so-called clear-sky bias (CSB) and is in the order of 10% (Sohn and Bennartz, 2008). Additionally, the majority of satellite-based TCWV CDRs rely on measurements from polar-orbiting satellites. Thus, observations of TCWV are only available at specific times of the day, and the full day is not covered with samples. This might cause a diurnal cycle sampling bias. Falk et al. (2022) utilised ERA5 data to assess the joint bias caused by the clear-sky nature of retrievals and the lack of full-day coverage. The basic approach was to compare full-day, all-sky averages with clear-sky averages at specific time slots using ERA5 output. Here their results are briefly summarised as follows: the overall average CSB is approximately -0.9 kg m$^{-2}$. The CSB is generally negative, with the largest negative values in the ITCZ and storm track regions. Regions with positive areas were observed over stratocumulus regions and Antarctica. Given the dependency on clouds and, among others, the movement of the ITCZ, the spatial distribution of the CSB is a function of season. An example comparison of the CSB at 10 local time and the CSB using the full diurnal information from ERA5 exhibits fairly similar results and mainly a noisier appearance. However, as shown for South America and over the full course of the day by Falk et al. (2022), a dependency on local time can be present at a regional scale. To some extent, the sign and the spatial features observed in Figure 7 for AIRWAVE, NNHIRS, UWHIRS and MODIS Terra IR agree with the features from Falk et al. (2022). This is also valid over the ocean for the other MODIS products. Thus, the predominant clear-sky sampling nature of these products might at least partly explain the observed features. The overall bias for AIRS (except over tropical South America) and CM SAF/WV_cci (except over central Eastern China, though potentially related to uncertainties arising from the treatment of aerosols) is fairly small while GOME Evolution (land and ocean), MPIC OMI (land and ocean), MODIS TERRA+AQUA NIR (land) and MODIS AQUA IR (land) exhibit positive biases, in contrast to the expectation. It is noted that the ensemble is dominated by all-sky data records from reanalysis and microwave observations. Nonetheless, the ensemble mean has members from visible, NIR and IR observations as well and, even more so, contains a mixture of information and uncertainty.

## 6 Conclusions

We introduce a new version of the G-VAP data archive. It features new versions and newly added data records while some are superseded or removed from the archive. The main change to the previous version is the extended temporal coverage from 1979 - 2019 and that the individual temporal coverage of the data records is kept. A flag was added to easily identify periods in which no data is available. Based on the updated G-VAP data archive, various analysis was carried out to characterise the individual data records, with a focus on their fitness to allow climate analysis. This includes the analysis of trends, breakpoints and climate variability. Overall results confirm previously achieved results in G-VAP (see Schröder et al. (2016) and Schröder et al. (2019)), though with differences in the details that can be expected given changes in the methodology. New results include the following: The intercomparison effort has been extended by analysing standard deviation, coefficient of variance and range between ensemble minimum and maximum. Associated results emphasise the large variability between data records over the poles, South East Asia and dry atmospheres in general. Despite the efforts of e.g. Crewell et al. (2021) dedicated regional

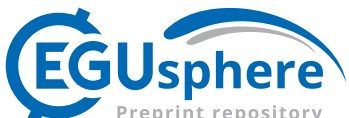

quality analysis of water vapour records is still needed. The ENSO contribution to the TCWV variability is largely consistent in terms of spatial patterns between the data records. However, it exhibits considerable variability in strength. An analysis of phase shifts per event and large-scale regions indicates the presence of teleconnections and, if carried out with a dedicated GEWEX Process Evaluation Study (PROES), might also provide explanations for this. Additional analysis of the clear-sky bias and to some extent of the diurnal sampling bias was carried out and confirms previous conclusions that such a bias significantly

impacts predominantly sampled clear-sky TCWV products if aggregated in space or time. Nonetheless, associated uncertainty estimation, additional analysis on potential all-sky versus cloudy-sky observations and how specific sampling impacts climate change estimates given orbital drift and climate change impact on dependent variables such as clouds and their diurnal cycle is still a wide field for additional scientific analysis. Finally, a careful improvement of the temporal stability of HIRS TCWV data records is highly beneficial, given, that HIRS offers the unique opportunity to retrieve TCWV over land and ocean beginning

in the late 1970s.

*Data availability.* All data records used in this study are freely available for scientific use, with details on the various sources given in the text.

**Appendix A: List of Acronyms**

Acronyms used in this study are defined here in Table A1.

**Appendix B: Climate indices**

Further details on climate indices used in this study are presented here in Table B1.

**Appendix C: Global trends for 'long' period**

The spatial pattern of global trends calculated for each record that covers the 'long' analysis period are shown here in Figure A1.

*Author contributions.* TT and MS collected the original TCWV data records used in version 2 of the G-VAP archive. TT processed all records per the common grid format described in this study. TT and MS analysed all the records within the archive and generated results. TT and MS wrote the first draft of the paper, with all authors contributing to the discussion and interpretations of results and manuscript revision. Data record PI's provided expert advice on interpretation of breakpoint analysis.



*Competing interests.* The contact author has declared that none of the authors has any competing interests

*Acknowledgements.* TT acknowledges the financial support of ESA (contract No. 4000131292) via the Living Planet Fellowship Program. MS acknowledges the financial support of the EUMETSAT member states through the Satellite Application Facility on Climate Monitoring (CM SAF). UF acknowledges the financial support of ESA (Contract No. 4000123554) via the Water_Vapour_cci project of ESA's Climate Change Initiative (CCI). The AIRWAVEv2 work has been performed under the ESA-ESRIN contract no. 4000108531/13/I-NB. Support for the Twentieth Century Reanalysis Project is provided by the U.S. Department of Energy, Office of Science Biological and Environmental

Research (BER), by the National Oceanic and Atmospheric Administration Climate Program Office, and by the NOAA Physical Sciences Laboratory. The AIRS project work was carried out at the Jet Propulsion Laboratory, California Institute of Technology, under a contract with the National Aeronautics and Space Administration (80NM0018D0004). JKN acknowledges the financial support of the EUMETSAT member states through the Radio Occultation Meteorology Satellite Application Facility. The EMiR data record was created under ESA contract No. 4000109537/13/I-AM. The NVAP-M data record was supported by NASA MEaSUREs. contract MEAS-06-0023. The MODIS

IR TPW products have been developed at the Space Science and Engineering Center, University of Wisconsin-Madison, funded by NASA (grant number NNX14AN48G). The HIRS UWisconsin data record was created under NOAA, contract No. NA15NES4320001.



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





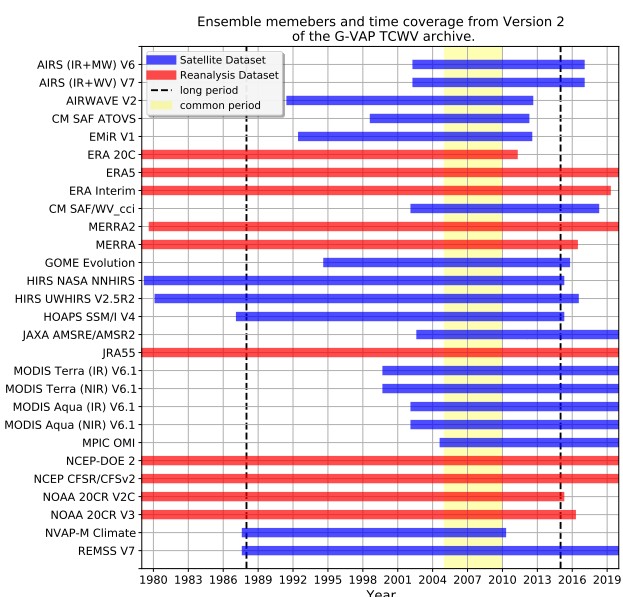

**Figure 1.** Coverage of TCWV records included in version 2 of the G-VAP archive from January $1^{st}$ 1979 to the $31^{st}$ of December 2019. Satellite records are shown in blue, whilst reanalysis is indicated in red. The two analysis periods, "long" (1988-01-01 to 2014-12-31) and "short" (2005-01-01 to 2009-12-31), are represented by black dashed lines and yellow shaded region, respectively.



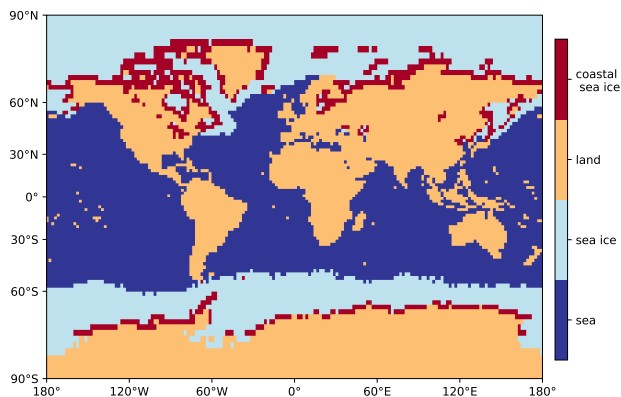

**Figure 2.** Conservative sea ice mask produced from a combination of the ESA CCI land cover classification and EUMETSAT OSI SAF sea ice concentration products. Sea ice or coastal sea ice is flagged if any common grid cell contains detected sea ice between 1988-2014.

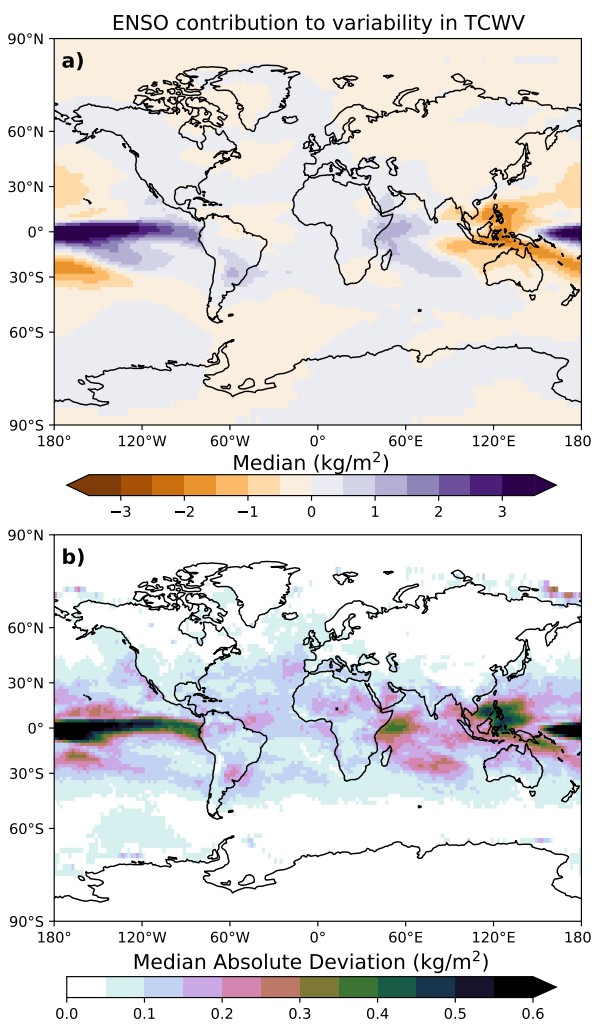

**Figure 3.** Archive ensemble median ENSO contribution to the variability in TCWV for the common long period (1988-2014) and median absolute deviation (MAD) to the ensemble median ENSO contribution.




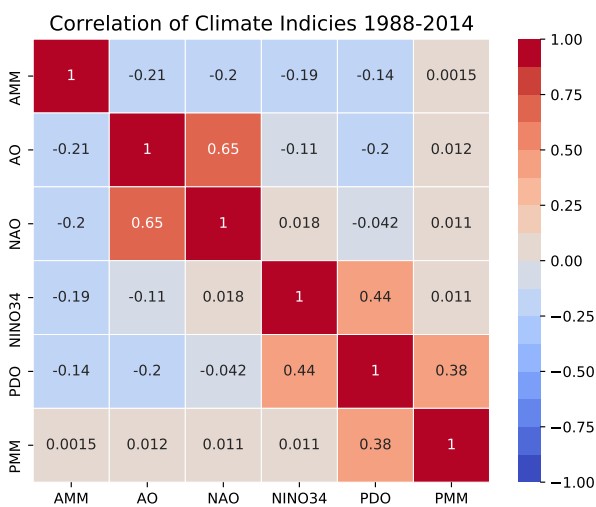

**Figure 4.** Correlation coefficients between the different climate indices used in this study for the period 1988 to 2014.



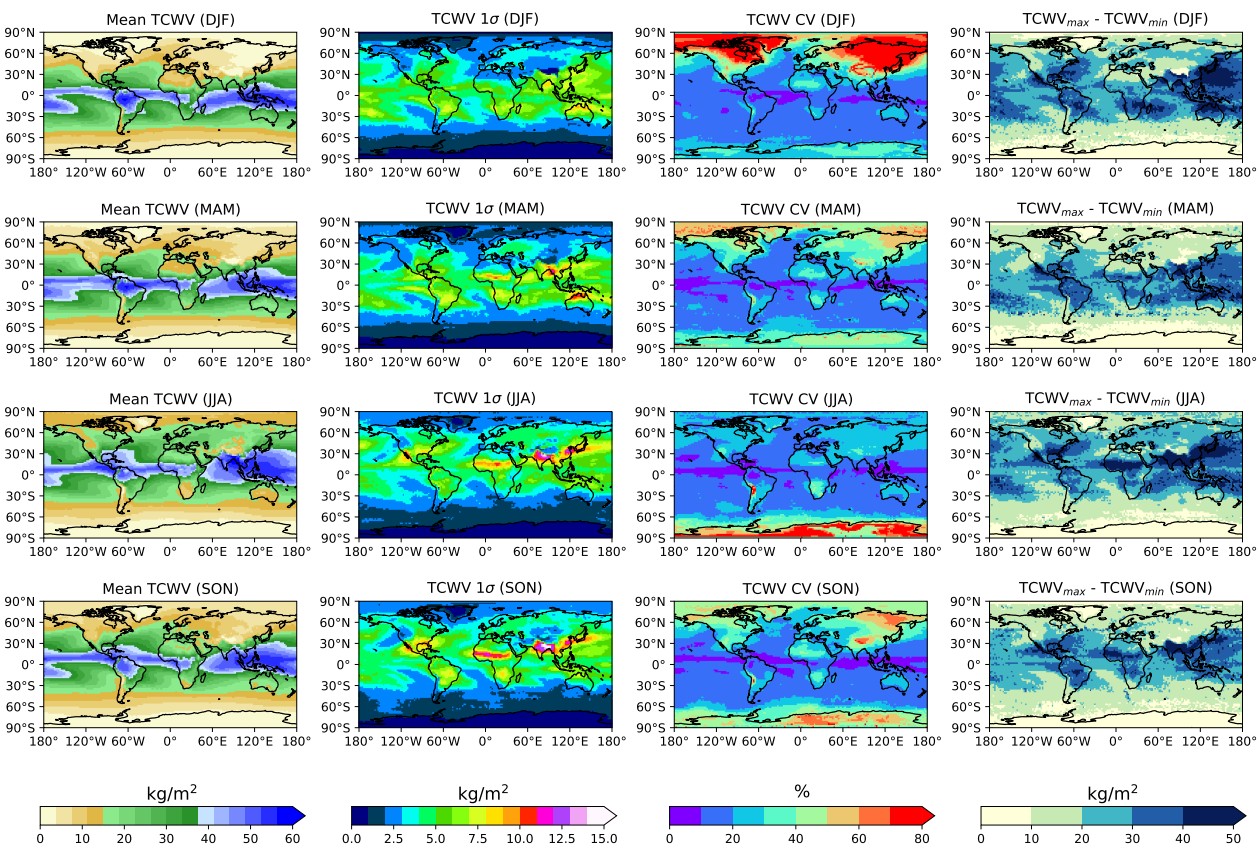

**Figure 5.** Seasonal maps of the mean archive ensemble total column water vapour (TCWV) over the common period 2005-2009. Also included is the standard deviation of the seasonal mean ($\sigma$), the respective coefficient of variance (CV), and the difference between the seasonal maximum and minimum TCWV values (TCWV$_{max}$-TCWV$_{min}$). While the largest (absolute) variability is seen in the Tropics, the greater relative difference (>50%) is found in polar regions.



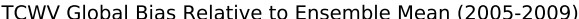

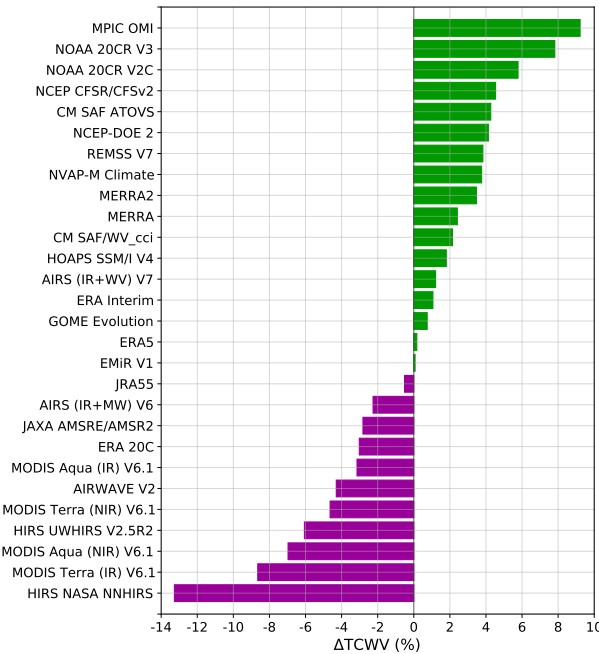

**Figure 6.** Global TCWV differences of each archive member to the ensemble mean for the common period 2005-2009. Average global differences have been summed and then normalised by the ensemble mean TCWV. For records that have missing data over land or polar regions, the ensemble mean is recalculated to account for differences, excluding these areas with missing data.



**Figure 7.** Ensemble member total column water vapour (TCWV) biases relative to the ensemble mean for the common period of all records (see Figure 1).



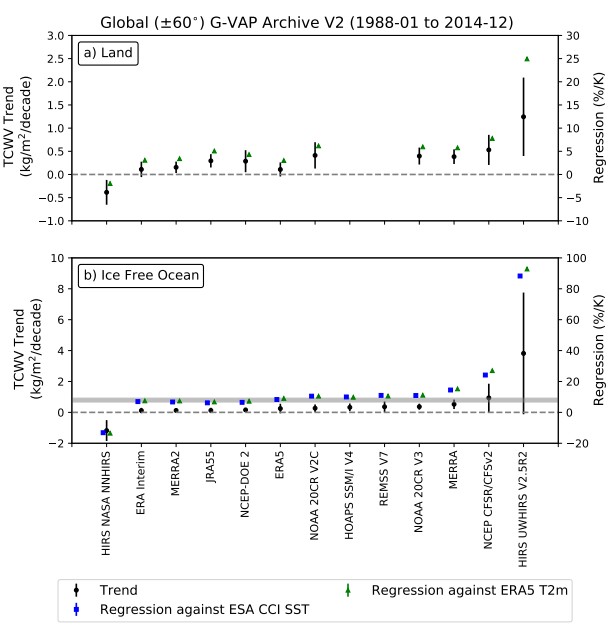

**Figure 8.** Trend estimates in total column water vapour (TCWV) in kg/m$^2$ per decade for data records that span the common 'long' period (1988-2014) for global land (a) and ice-free oceans (b) between $\pm 60°$. Trends are represented as black dots with the estimated uncertainty of the trend shown in vertical bars. Also shown on the right-hand y-axes are the regression coefficient (%/K) for each data record. Regression has been carried out against sea surface temperature (SST) from the ESA Climate Change Initiative (CCI) and 2-metre air temperature (T2m) from ERA5. These are shown as blue squares and green triangles, respectively, with the regression error as vertical bars. The grey-shaded region on the bottom figure denotes the expected range of regression values and actual expectation based on the mean change in SST (Wentz and Schabel, 2000)

.





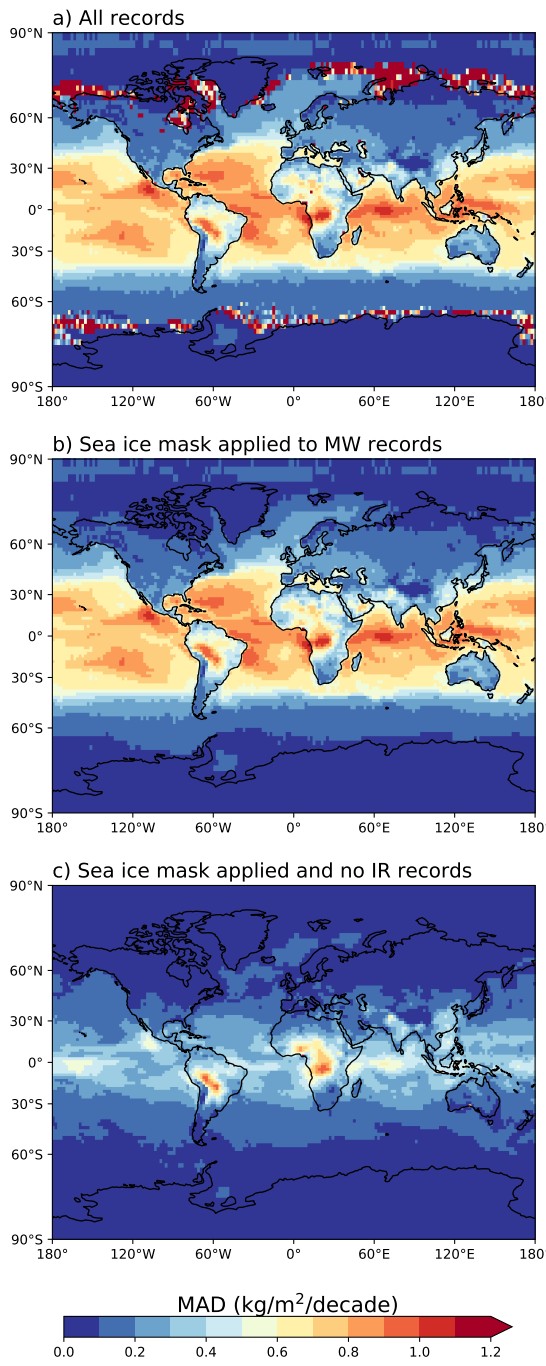

**Figure 9.** Mean absolute deviation (MAD) of archive total column water vapour (TCWV) trend estimates for the period between 1988-2014 (a). High MAD values observed in the Tropics are on the order 25% of observed trends, while the highest MAD values (>1.2 kg/m$^2$/decade) are observed at high latitudes and are related to sea ice. Application of the sea-ice mask (see Figure **??**) to the microwave (MW) records removes these effects seen at high latitudes (b), while the high variability seen in the tropics is related to the Infrared (IR) only products (c).



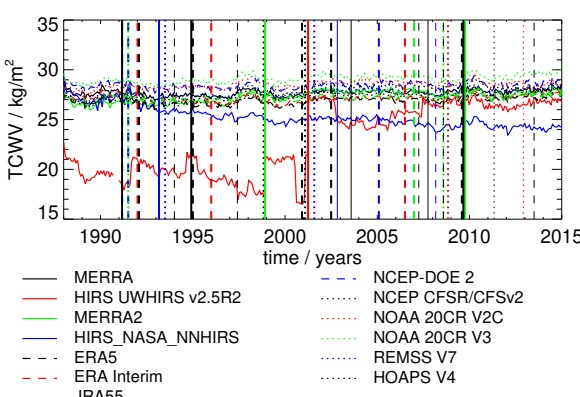

**Figure 10.** Anomaly time series of TCWV, averaged over the global ice-free ocean within 60°N/S, from members of the G-VAP data archive that cover the period 1988-2014. Each average of the TCWV time series was added to each anomaly time series in order to visualise biases between the data records. Vertical lines denote observed breakpoints using the same colour bar as for the anomaly time series. The vertical line is plotted in bold in case the PMF and SNH tests both detected the breakpoint.



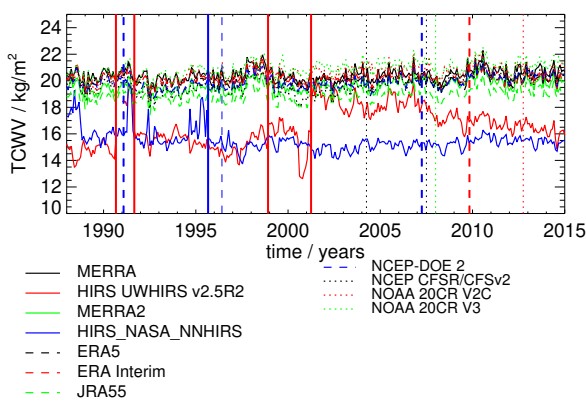

**Figure 11.** As Figure 10 but for global land regions within 60°N/S.





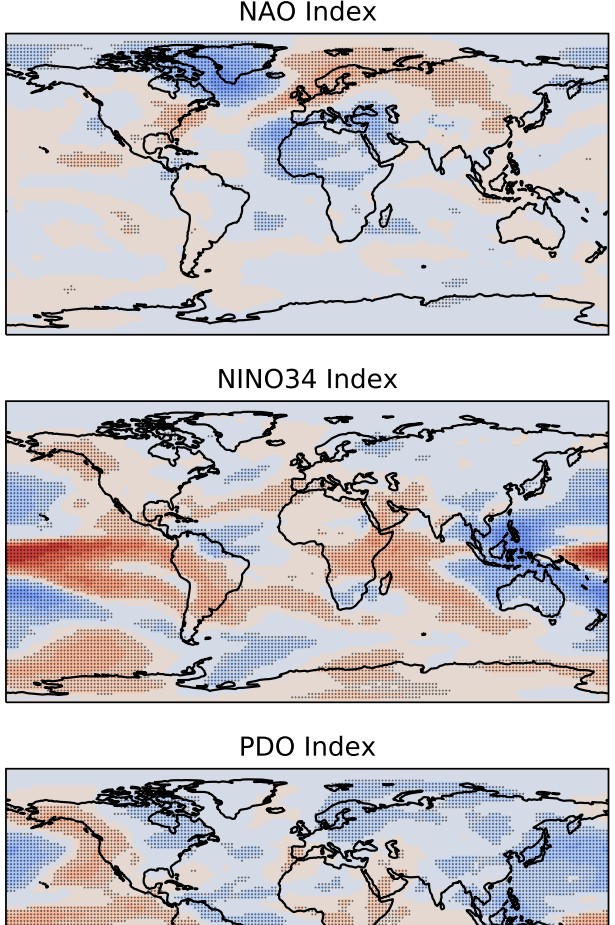

**Figure 12.** Global maps of the correlation between ERA5 total column water vapour (TCWV) and the North Atlantic Oscillation (NAO), El Niño Southern Oscillation Index 3.4 (NINO3.4), Pacific Decadal Oscillation (PDO) climate indices. The stippling indicates areas where the correlation is within the 95% confidence level.



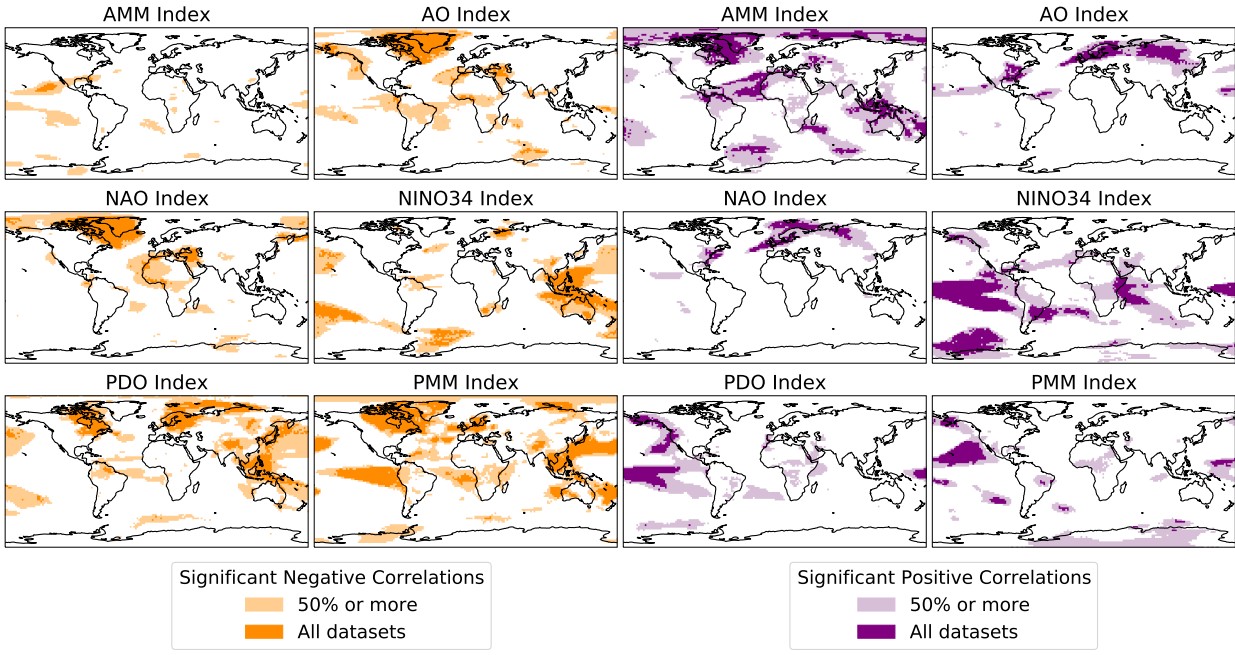

**Figure 13.** Summary of global regions where the water vapour records that cover the long period (1988-2014) significantly correlate with the different climate indices. Negative correlations are shown in orange on the left, and positive correlations are shown in purple on the right. Areas coloured in the lighter shade of colour indicate where at least 50% of all water vapour datasets show significant correlations, while darker-coloured areas indicate where 100% of records show significant correlations to the climate indices.



**Figure A1.** Global maps of total column water vapour (TCWV) trend estimates from all records between 1988 and 2014. Regions where trends were not statistically significant are highlighted with stippling. These regions tend to coincide with stronger trends; however, it should be noted that the reported trends by G-VAP are used to identify issues rather than an analysis of climate change. Therefore, it is more likely these areas experience higher ranges of variability, which introduces noise into the calculation (Weatherhead et al., 1998) Thus, longer time series are required to improve confidence in reported trends.



**Table 1.** Details of data records from version 1 of the G-VAP archive and their status within the version 2 release. Version 1 of the archive is available from https://doi.org/10.5676/EUM_SAF_CM/G-VAP/V001.

|   | Data record | status |
|---|---|---|
| 1 | AIRWAVE | superseded by version 2 (Castelli et al., 2019)* |
| 2 | ATOVS CM SAF | extended coverage |
| 3 | AMSR-E JAXA | extended coverage and inclusion of AMSR2 |
| 4 | AMSR-E REMSS | not used |
| 5 | EMiR | extended coverage |
| 6 | ERA-Interim | extended coverage |
| 7 | ERA-20C | extended coverage |
| 8 | GOME/SCIAMACHY/GOME-2 GlobVapour | superseded by new product |
| 9 | HIRS UWisconsin | extended coverage |
| 10 | HOAPS SSM/I V3 | superseded by version 4 |
| 11 | JRA-55 | extended coverage |
| 12 | Merged Microwave REMSS | extended coverage |
| 13 | MERIS GlobVapour | not used |
| 14 | MERRA | extended coverage |
| 15 | MERRA-2 | extended coverage |
| 16 | MODIS NIR & IR Aqua NASA (MYD08_M3**) | extended coverage and inclusion of MODIS Terra (MOD08_M3) and IR products from both Aqua & Terra |
| 17 | NCEP CFSR | extended using CFSv2 |
| 18 | nnHIRS | extended coverage |
| 19 | NVAP-M Climate | extended coverage |
| 20 | NVAP-M Ocean | not used |
| 21 | SSM/I+MERIS GlobVapour | superseded by new product |
| 22 | TMI REMSS | droppped |

* The AIRWAVEv2 data are available on request from the following authors: Elisa Castelli (e.castelli@isac.cnr.it), Enzo Papandrea (e.papandrea@isac.cnr.it). ** Only NIR TCWV water vapour used in version 1 of the archive



**Table 2.** Break points detected by the PMF and SNH tests and coincident events. Results are shown for the global ocean. The breakpoints are characterised by the time of the event (yyyy-mm) and their strength in kg m$^{-2}$. The strength of the breakpoint is printed in bold if both tests agree on the date within ±3 months. Break events marked with a "*" are taken from Schröder et al. (2019). Information on satellite status is taken from https://www.ospo.noaa.gov/Operations/POES/status.html.

| Data record | Date PMF | Breaksize PMF | SNH | Event |
|---|---|---|---|---|
| **global ocean** | | | | |
| ERA-Interim | 1991-12 | **-0.48** | **-0.50** | Approximate i) end of assimilation of F08 data, ii) start of assimilation of F10 data, and iii) end of assimilation of NOAA10 [1]* |
| | 1995-12 | **-0.24** | **-0.14** | Approximate start of assimilation of F13 data [1] |
| | 2006-06 | 0.18 | | Approximate i) end of assimilation of F15 and NOAA14 data, ii) change from GOES10 to GOES11, and iii) start of assimilation of Meteosat 5 and 8 data, see [1], see text* |
| ERA5 | 1991-02 | -0.22 | | Approximate start of assimilation of F10 data |
| | 1992-01 | **-0.22** | **-0.35** | Approximate start of assimilation of F11 data [2] |
| | 1993-12 | 0.10 | | Approximate end of assimilation of F11 data [2] |
| | 1994-12 | -0.27 | | Approximate i) end of assimilation of NOAA11 data, and ii) start of assimilation of F11 (after a 1 year interruption) [2] |
| | 1997-05 | -0.11 | | Approximate i) start of assimilation of NOAA11 data, after a 2.5 year interruption, and ii) start/end of assimilation of F14/F10 [2] |
| | 2000-11 | 0.12 | | Approximate start of assimilation of NOAA16 data [2] |
| | 2002-06 | 0.12 | | Approximate start of assimilation of NOAA17 data [2] |
| | 2007-03 | 0.09 | | Approximate start of assimilation of Metop-A and Meteosat9 data [2] |
| | 2008-07 | -0.13 | | Unclear |
| | 2009-07 | -0.16 | | Approximate start of assimilation of NOAA19 and F17 data [2] |
| | 2013-06 | -0.09 | | Approximate i) start of assimilation of FY-3B data, and ii) end of assimilation of NOAA19 data |
| HIRS UWisconsin | 2001-03 | **6.14** | **8.87** | NOAA16 operational since 2001-03 [2]. |
| JRA55 | 2006-12 | **0.22** | **0.16** | Launch of Metop-A in October 2006. Approximate end of assimilation of monthly surface meteorological data in China [3], see text |
| MERRA | 1998-11 | **0.54** | **0.53** | Start of assimilation of NOAA15 in July 1998. Note: assimilation of AMSU-A & AMSU-B (NOAA15) started on 1998-11-01, while assimilation of HIRS data (NOAA15) began on 1998-07-02 [4]* |
| | 2009-09 | -0.47 | | Approximate end of assimilation of F13 data |

references: [1] Dee et al. (2011); [2] Hersbach et al. (2020); [3] Kobayashi et al. (2015); [4] Rienecker et al. (2011); [5] McCarty et al. (2016); [6] Gelaro et al. (2017); [7] Saha et al. (2010); [8] Schröder et al. (2016); [9] Slivinski et al. (2019)





**Table 2.** Table 2 continued

| Data record | Date | Breaksize | | Event |
|---|---|---|---|---|
| | PMF | PMF | SNH | |
| **global ocean** | | | | |
| MERRA2 | 1991-02 | **-0.31** | **-0.33** | Start of assimilation of F10 data on 1990-12-09 [5]* |
| | 1994-11 | **-0.11** | **-0.15** | Approximate end of assimilation of NOAA11 data [5] |
| | 2003-07 | **0.13** | **0.09** | No obvious coincidence with a change in space-based water vapour sensors; approximate end of assimilation of data from AMSU-A on NOAA17; strong drop in assimilated wind observations from ERS2 in June 2003 [5] |
| | 2007-09 | **0.13** | **0.10** | Approximate start of assimilation of surface wind from WindSat in 2007-08, strong increase in the number of assimilated AMVs from JMA and a decrease in the number of assimilated AMVs from MODIS [5,6]* |
| | 2009-08 | **-0.21** | **-0.22** | Approximate start and end of assimilation from various satellite data in April, November & December 2009; end of assimilation of rain rates from SSM/I in September 2009 [5] |
| NCEP CFSR/CFSv2 | 1998-10 | **1.07** | **1.20** | Approximate start of i) assimilation of NOAA15 data, and end of assimilation of NOAA11 & NOAA14; change from assimilating GOES09 to GOES10 [7]* |
| | 2001-01 | **0.46** | **0.33** | Approximate start of assimilation of data from NOAA16 [7] |
| | 2011-04 | -0.32 | | Unclear |
| NCEP-DOE 2 | 1991-06 | **-0.26** | **-0.32** | Unclear |
| | 2005-06 | 0.29 | | Unclear |
| | 2006-12 | **0.24** | **0.21** | Unclear |
| | 2008-02 | **-0.34** | **-0.42** | Unclear |
| NNHIRS | 1993-02 | **-0.88** | **-0.52** | See [8] for discussion on the results related to NVAP-M. It seems that nnHIRS also exhibits increased uncertainties then.* |
| | 2002-10 | 0.52 | | NOAA-17 is operational since 2002-10 |
| NOAA 20CR V2C | 2008-10 | -0.28 | | Unclear |
| | 2012-11 | **-0.48** | **-0.39** | Approximate change in the source of in-situ SST data [9] |
| NOAA 20CR V3 | 1991-06 | **-0.24** | **-0.25** | Unclear |
| | 2008-07 | -0.31 | | Unclear |
| REMSS V7 | 1993-06 | **0.24** | **0.11** | See [8]* |
| | 2001-07 | **0.14** | **0.06** | see text |

references: [1] Dee et al. (2011); [2] Hersbach et al. (2020); [3] Kobayashi et al. (2015); [4] Rienecker et al. (2011); [5] McCarty et al. (2016); [6] Gelaro et al. (2017); [7] Saha et al. (2010); [8] Schröder et al. (2016); [9] Slivinski et al. (2019)



**Table 3.** As Table 2 but for global land surfaces. For the following data records, no breakpoints were detected over land: JRA55, MERRA, and MERRA2.

| Data record | Date | Breaksize | | Event |
|---|---|---|---|---|
| | PMF | PMF | SNH | |
| **global land** | | | | |
| ERA-Interim | 2009-10 | **-0.23** | **-0.18** | Approximate start of assimilation of AMSR-E and F16 data, see [1] |
| HIRS UWisconsin | 1990-08 | **3.86** | **3.42** | HIRS/2 on NOAA-10 became very noisy. Early HIRS/2 data is an outlier because of the missing split window (12 $\mu$m) band. |
| | 1991-08 | **-1.74** | **-2.03** | NOAA12 is operational since 1991-09. |
| | 1998-11 | **1.63** | **1.65** | NOAA15 is operational since 1998-12. |
| | 2001-03 | **3.48** | **4.82** | NOAA16 operational since 2001-03. |
| NCEP CFSR/CFSv2 | 2004-03 | **0.52** | **0.41** | Approximate start of assimilation of data from AIRS and AMSU-A onboard MODIS-Aqua [7]. |
| | 2007-06 | **0.47** | **0.40** | Approximate start of assimilation of data from Metop-A [7]. |
| NCEP-DOE 2 | 1991-01 | **-0.2** | **-0.14** | Unclear |
| | 1996-05 | 0.16 | | Approximate date of new snow climatology. |
| | 2007-03 | **0.30** | **0.25** | Unclear |
| NNHIRS | 1995-08 | -1.15 | | The break does not seem to coincide with a change in NOAA satellites. It marks the end of a period of approximately two years with a series of anomalies. |
| NOAA 20CR V2C | 2012-09 | -0.60 | | Unclear |
| NOAA 20CR V3 | 2007-12 | -0.43 | | Unclear |

references: [1] Dee et al. (2011); [2] Hersbach et al. (2020); [3] Kobayashi et al. (2015); [4] Rienecker et al. (2011); [5] McCarty et al. (2016); [6] Gelaro et al. (2017); [7] Saha et al. (2010); [8] Schröder et al. (2016); [9] Slivinski et al. (2019)



**Table A1.** Overview of utilised abbreviations and their definitions

| Abbreviation | Definition |
| --- | --- |
| AIRS | Atmospheric Infrared Sounder |
| AIRWAVE | Advanced InfraRed Water Vapour Estimator |
| AMSR2 | Advanced Microwave Scanning Radiometer 2 |
| AMSR-E | Advanced Microwave Scanning Radiometer for EOS |
| AMIP | Atmospheric Model Intercomparison Project |
| AR6 | Sixth Assessment Report |
| ATOVS | Advanced TIROS Operational Vertical Sounder |
| CFSR | Climate Forecast System Reanalysis |
| CFSv2 | Coupled Forecast System model Version 2 |
| CM SAF | Satellite Application Facility on Climate Monitoring |
| ECMWF | European Centre for Medium-Range Weather Forecasts |
| ENSO | El Niño Southern Oscillation |
| EMiR | ERS/Envisat MWR Recalibration and Water Vapour TDR Generation |
| ERA-Interim | ECMWF Interim Reanalysis |
| ERA-20C | ECMWF $20^{th}$ Century Reanalysis |
| ERA5 | ECMWF Reanalysis v5 |
| ESA | European Space Agency |
| EUMETSAT | European Organisation for the Exploitation of Meteorological Satellites |
| GDAP | Data and Assessments Panel |
| GEWEX | Global Energy and Water cycle Exchanges |
| GOME/GOME-2 | Global Ozone Monitoring Experiment |
| GOME EVOL | GOME Evolution |
| G-VAP | GEWEX Water Vapor Assessment |
| HIRS | High-Resolution Infrared Sounder |
| HOAPS | Hamburg Ocean Atmosphere Parameters and Fluxes from Satellite data |
| JAXA | Japan Aerospace Exploration Agency |
| JMA | Japan Meteorological Agency |
| JRA-55 | Japanese 55-year Reanalysis |
| MAD | Median absolute deviation |
| MERIS | Medium Resolution Imaging Spectrometer |
| MERRA | Modern-Era Retrospective analysis for Research and Applications |



**Table A1.** Table A1 continued.

| Abbreviation | Definition |
| --- | --- |
| MERRA-2 | MERRA Version 2 |
| MODIS | Moderate-resolution Imaging Spectroradiometer |
| MPIC | Max Planck Institute for Chemistry |
| NASA | National Aeronautics and Space Administration |
| NCEP | National Centers for Environmental Prediction |
| NIR | Near-infrared |
| nnHIRS | |
| NOAA | National Oceanic and Atmospheric Administration |
| NVAP | NASA Water Vapor Project |
| NVAP-M | NVAP – Making Earth Science Data Records for Research Environments |
| MW | Microwave |
| OLCI | Ocean and Land Colour Instrument |
| OMI | Ozone Monitoring Instrument |
| PROES | process evaluation studies |
| REMSS | Remote Sensing Systems |
| SCIAMACHY | SCanning Imaging Absorption spectroMeter for Atmospheric CartograpHY |
| SSM/I | Special Sensor Microwave Imager |
| SSMIS | Special Sensor Microwave Imager/Sounder |
| TCWV | Total Column Water Vapour |
| TIROS | Television Infrared Observation Satellite |
| TMI | Tropical Rainfall Measuring Mission's Microwave Imager |
| WCRP | World Climate Research Panel |
| WV_cci | Water Vapour Climate Change Initiative |



**Table B1.** Climate indices used in this study.

| Index | Description | Reference | Source |
|-------|-------------|-----------|--------|
| AMM | Atlantic Meridional Mode. The AMM spatial pattern is defined via applying Maximum Covariance Analysis (MCA) to the sea surface temperature (SST) and the 10 m wind field over the time period 1950-2005 over the region 21°S to 32°N and 74°W to 15°E. | Chiang and Vimont (2004) | http://www.esrl.noaa.gov/psd/data/timeseries/monthly/AMM/ |
| AO | Arctic Oscillation. The daily AO index is constructed by projecting the daily (00Z) 1000 hPa height anomalies pole ward of 20°N onto the loading pattern of the AO. | Thompson and Wallace (1998) | http://www.cpc.ncep.noaa.gov/products/precip/CWlink/daily_ao_index/ao.shtml |
| NAO | North Atlantic Oscillation.The principal component (PC)-based indices of the NAO are the time series of the leading EOF of sea level pressure anomalies over the Atlantic sector, 20°-80°N, 90°W-40°E. | Walker (1924), Rogers (1984), Barnston and Livezey (1987), & Hurrell (1995) | http://www.esrl.noaa.gov/psd/data/climateindices/list/ |
| NINO34 | El Nino Southern Oscillation Index 3.4. Average SST anomaly in the region between 5°N and 5°S as well as 170°W and 120°W. | Walker (1924) & Rasmusson and Carpenter (1982) | http://www.esrl.noaa.gov/psd/data/climateindices/list/ |
| PDO | Pacific Decadal Oscillation. The PDO is defined as the leading principal component of North Pacific monthly sea surface temperature variability (poleward of 20N for the 1900-93 period). | Mantua et al. (1997) & Zhang et al. (1997) | http://jisao.washington.edu/pdo/ |
| PMM | Pacific Meridional Mode. Like AMM, the PMM pattern is defined via applying MCA to SST and the zonal and meridional components of the 10 m wind field for period 1950-2005 over the region (21°S-32°N, 74°W-15°E | Chiang and Vimont (2004) | https://psl.noaa.gov/data/timeseries/monthly/PMM/ |