# Peer review of "Evaluation of Total Column Water Vapour Products from Satellite Observations and Reanalyses within the GEWEX Water Vapor Assessment"

_EGUsphere, 2023_

## Author Comment (AC1)

**Authors' Response to Reviews of**

**Evaluation of Total Column Water Vapour Products from Satellite Observations and Reanalyses within the GEWEX Water Vapor Assessment**

Tim Trent, Marc Schroeder, Shu-Peng Ho, Steffen Beirle, Ralf Bennartz, Eva Borbas, Christian Borger, Helene Brogniez, Xavier Calbet, Elisa Castelli, Gilbert P. Compo, Wesley Ebisuzaki, Ulrike Falk, Frank Fell, John Forsythe, Hans Hersbach, Misako Kachi, Shinya Kobayashi, Robert E. Kursinsk, Diego Loyola, Zhengzao Luo, Johannes K. Nielsen, Enzo Papandrea, Laurence Picon, Rene Preusker, Anthony Reale, Lei Shi, Laura Slivinski, Joao Teixeira, Tom Vonder Haar, and Thomas Wagner

EGUsphere, https://doi.org/10.5194/egusphere-2023-28087, 2023

Please find below our responses to Reviewer 1, We include the original comments in black, our comments in green, and any alterations to text in blue.

This study does an impressive job of consolidating the information in a wide variety of data sets, and manages to nicely balance depth of analysis with breadth of topic. The results are relevant and interesting, and reveal several important insights. For these reasons, the study is appropriate for publication. I recommend some minor changes to the text.

The first recommended change is to revamp the discussion of diurnal cycle in around line 465 and elsewhere. The text mentions potential biases from a lack of 'full day' observations without being specific about what this means. Sun-synchronous satellite measurements of the true diurnal cycle (exactly 24 hour period) taken every 12 hours will average to the diurnal mean. Consequently, the long-term mean will be unbiased. In contrast, sampling once daily will introduce a bias since the average is over only a single phase of the diurnal cycle. This is the case for the daytime-only satellite observations. The distinction between these two situations isn't clear from the text, but some minor editing should fix this. (A semidiurnal cycle in TCWV or the effects of diurnally varying clouds on sampling further complicate the picture. This is worth mentioning, but beyond the scope of the study.) Because of the potential for the diurnal cycle to introduce a bias, the study should mention the times of day of all satellite data sets and whether the observations are obtained only during day or night. Some information about orbit local times and day-only sampling is provided in the current text, but not for every data set. Consistent statements for all satellite data sets would be helpful. A table may even be appropriate, though that is at the authors' discretion.

Indeed, it is helpful to add information on temporal sampling characteristics. We propose to add the following (section 5, after the reference to Sohn and Bennartz, 2008):

Also, TCWV retrieved from observations in the visible and NIR rely on reflected solar radiation and thus, is available during day time only (affected products are: CM SAF/WV_cci

over land, GOME Evolution, MODIS (NIR), and MPI-C OMI). Additionally, the majority of satellite-based TCWV CDRs rely on measurements from polar-orbiting satellites. Thus, observations of TCWV are only available at specific times of the day, and the full day is not covered with samples. The following data records rely on single sensor observations only, with the equator crossing time given in brackets: AIRS v6/V7 (10:30+13:30), AIRWAVE v2 (10, 10:30), CM SAF/WV_cci (10:30, over land), EMiR (10, 10:30), GOME Evolution (10, 10:30), MODIS Terra (10:30), MODIS Aqua (13:30), and MPI-C OMI (13:40). All other satellite based data records sample more frequently, but at different times and frequencies, partly affected by orbital drift and partly varying in time. Details on satellite equator crossing times can be found at https://space.oscar.wmo.int/satellites. Examples of equator crossing times affected by orbital drift can be found at: STAR - Global Vegetation Health Products: Equatorial Crossing Time (ECT) for NOAA Polar Satellites and Remote Sensing Systems (remss.com). This specific sampling might cause a diurnal cycle sampling bias.

Following the above discussions by the review we added (after the reference to Falk et al., 2022):

It is noted that the effect of orbital drift, of potential changes in diurnal cycles of TCWV and clouds and of potential changes in spatial cloud distributions with climate change have not been considered here.

A discussion of HOAPS data set is needed, especially since it is the reference data set over ocean. (The first reference on p. 9 does not even spell out the acronym.) Even a short paragraph would be very helpful.

We agree and added a short introduction of HOAPS to section 3.3:

As ERA5 was already introduced in section \ref{era5} we briefly recall here key characteristics of HOAPS. HOAPS is a product suite of satellite based climate data records, including TCWV, over the global ice free oceans. TCWV is derived from quality controlled, recalibrated and intercalibrated measurements from SSM/I and SSMIS passive microwave radiometers (Fennig et al., 2020), except for the SST, which is taken from AVHRR measurements. TCWV is retrieved with a 1D-Var scheme. The data record covers the time period from July 1987 to December 2014 and has global coverage, i.e., within ±180° longitude and ±80° latitude. The product is available as monthly averages and 6-hourly composites on a regular latitude/longitude grid with a spatial resolution of 0.5° x 0.5° degrees.

The text would also benefit from a brief discussion of the use of ocean surface temperature and near-surface air temperature over land in calculating the temperature-TCWV regression. The use of different reference temperatures has implications for interpreting the temperature-humidity relationship over land versus over ocean, and intercomparison of land and ocean relationships.

The contrast in expectations is discussed in section 5. We expanded the discussions in section 5 by adding: "Following discussions in \cite{falk2022} on land areas, the relation between air temperature and surface temperature is complex and locally the difference between air and surface temperature reaches a few Kelvin. This depends on various factors, such as local time, cloudiness and surface type (e.g. \cite{good} and \cite{rayner})."

Here are more specific comments:

Line 38 forward.  How are these goals different from the first phase of the GVAP assessment?

This is indeed not very clear. We propose to change the first sentence after the list into: "These objectives are similar to the objectives from the first phase by only enhancing efforts directed towards process evaluation studies and regional analysis. In particular, as in phase 1 of G-VAP, the assessment effort focuses… "

Line 70.  It's not obvious that Aqua is included in this study.  Words like "now includes both Terra and the Aqua" would be helpful.

We propose to change the sentence into "In the case of the Moderate-resolution Imaging Spectroradiometer (MODIS) products, we now include the Terra and Aqua versions, i.e., MOD08\_M3 (new) and MYD08\_M3 (as in previous archive), respectively.".

Section 2.1.1.  The local time of the Aqua spacecraft is needed here.  Also, the reference to Manning et al. on line 83 should be Susskind et al.  Here is the first link on a web search of the title: https://docserver.gesdisc.eosdis.nasa.gov/public/project/AIRS/L2_ATBD.pdf

We thank the reviewer for spotting this error. Google Scholar had created the reference Manning et al. for the document linked in the comment above. Therefore, we have updated this appropriately so that the references are now Susskind et al. (2003, 2020) in the text. The local overpass times (13:30 and 01:30 hrs) have been added to the text.

Sections 2.1.2.  Please mention the local time of the satellite observations, and/or the drifting orbits, as appropriate.

We included the equator crossing times for satellite missions over land and a link to a webpage where equator crossing times of missions with microwave imagers onboard are shown.

Section 2.1.3.  Are the GOME observations obtained day and night, or 10:00 daytime only and not 22:00?

The GOME Evolution Climate product is a daytime-only product as they use the red part of the visible spectrum for retrievals from each instrument. We have made a slight edit to the text to ensure this is clear:

"The 'GOME evolution climate' product was generated within the GOME-Evolution project funded by ESA, and the retrieval is described in detail in Beirle et al. (2018). It is based on measurements from the satellite instruments Global Ozone Monitoring Experiment (GOME), Scanning Imaging Absorption Spectrometer for Atmospheric Chartography (SCIAMACHY), and GOME-2 in the red part of the visible spectral range, using the retrieval proposed in Wagner et al. (2003, 2006), with all satellites measurements (daytime only) occurring around 10:00 hours local time."

Section 2.1.4.  Some mention is needed of which satellite data sets considered in the study are also assimilated into ERA5.  Even a statement like "nearly all" would be helpful.

We propose to insert in line 132: "In fact, ERA5 assimilates most of the satellite measurements considered in this study (see Hersbach et al. 2020, table 4 and figure 5)."

Section 2.1.5.  The local times of Terra and Aqua overpasses are needed, and it should be stated that the retrievals are available both day and night (if so).

We added "this product uses the Thermal Infrared (TIR) bands 25 and 27 through 36 to retrieve temperature and moisture profiles, total-ozone burden, atmospheric stability, and atmospheric water vapour for daytime (10:30, 13:30 hrs local time) and night-time (22:30, 01:30 hrs local time) overpasses (Terra/Aqua respectively). The level 2 (L2) product contains the geophysical parameters at a resolution of 5x5 km for both clear-sky day & night scenes."

Section 2.1.6. Are MPIC daytime-only observations?

Yes, we updated the text to clarify "satellite with an equator crossing time of about 13:30 local time (daytime only observations)"

Line 175 forward.  Is the lack of SSM/I etc., data different from ERA5?  That seems to be the major distinction, but it's not stated.

We added ", in contrast to e.g. ERA5 which assimilates radiances from these sensors".

Line 195.  Is this bilinear interpolation in latitude and longitude.  This is worth mentioning because bilinear interpolation is actually a quadratic function, so has a specific (if confusing) definition.

We use linear spline interpolation to calculate the new grid centre values in this study. The text has been updated to clarify this:

"For the reanalysis records, each monthly gridded TCWV global field was first shifted in longitude space to run between -180∘ to 180∘ before being interpolated onto the centres of the archive common grid using a linear spline function"

Line 224.  This paragraph on comparison temperatures is worthy of its own short subsection.

We agree that the details regarding comparison to surface temperatures should be in their own subsection. We have added a heading and updated the text to read (plus the additional discussions mentioned above):

"3.3 Regression against Surface Temperatures

In addition to calculating trends, and to be consistent with Phase 1 of G-VAP, a regression of each TCWV record against surface temperature dataset(s) is performed following the approaches outlined in Dessler and Davis (2010) and Mears et al. (2007). If we assume relative humidity (RH) is constant, then the Clausius-Clapeyron relationship produces a ratio between changes in water vapour and temperature that is only dependent on temperature. Therefore, under constant RH and pressure assumptions, changes in water vapour mixing ratios can be transferred to saturation vapour pressure values. For a temperature change of 1 K, the expected change in mixing ratio is between 6% at 300 K and 7.5% at 275 K. These values then provide the limits of the range of expected regression coefficients against the chosen surface temperature data records used in this study:

- Over ocean sea surface temperature (SST) data from the European Space Agency (ESA) Climate Change Initiative (CCI) (Merchant et al., 2019; Merchant and Embury, 2020).

• For both land and ocean surfaces, surface air temperature (T2m) from the European Centre for Medium-Range Weather Forecasts (ECMWF) ERA5 reanalysis (Hersbach et al., 2020) is used.

Both temperature data sets were processed on the same grid as the TCWV records in the G-VAP archive for consistency"

Line 237.  As discussed above, this is the first mention of HOAPS.

See comment above.

Line 284.  The region described as California is almost entirely along the coast of Mexico. (The Baja California Peninsula is in the northern part of the region of interest, but is within Mexico and mostly outside of the California climate region.)

This is true and we changed the text accordingly.

Line 305.  Is this the first use of Delta-TCWV?  It does not appear to be defined earlier (or else I can't find it).

It is indeed only indirectly defined in the legends of Figures 6 and 7, i.e., it is the bias relative to the ensemble mean. We propose to define it at first occurrence in the text and in the figure captions (see updated manuscript).

Line 317.  We noted the wet biases in stratiform regions in Fetzer et al. (2006) and attributed them to a combination of subsidence-induced extensive cloudiness and low TCWV, while clearer conditions have higher TCWV.  Interestingly, that study considered only a few weeks of observations.

Thanks for this reminder. We included a reference to this paper also here.

Line 339.  Fix Figure ??.

Now Figure 2 is correctly referenced.

Line 347.  The 'also' can be deleted since it's redundant with 'In addition'.  A similar argument can be made for 'also' and its relationship to 'include' in the next sentence.

We deleted "also" in both sentences.

Line 352.  The sentence starting 'The UWHIRS…' should start a new paragraph since the HIRS data sets obviously deviate from the other data sets.  This is worth be highlighting.

We started a new paragraph in the updated manuscript. Consequently, the sentence starting "For the other products,..." also starts a new paragraph now.

Line 358.  Change "range of regression coefficients are" to "…coefficients is" since 'range' is the subject.

Changed as proposed.

Line 361.  Is the 'theoretical range' the one set by the Clausius-Clapeyron relationship?  It's not obvious from the text.

We added "following Clausius-Clapeyron".

Line 365.  Should "noise" be "large, and likely anomalous, high variability" or something like that?  A noise mechanism hasn't been clearly established.

Indeed, this should be formulated differently. We propose the following:

"Sea-ice boundaries show up as large MAD values with high spatial variability at high latitudes."

Line 388.  Change to "Also noteworthy are…"

Implemented as proposed.

Line 405.  Change the sentence ending slightly to "…trend estimates discussed earlier."

Implemented as proposed.

Line 411.  That should be "at least 50% and 100%" since 'or' implies either one or the other, but not both.

Changed as proposed.

Line 415.  Are the "expected regions" those mentioned in Table B1?  The text is ambiguous here.

Yes, we have updated the text to clarify this to the reader:

"Generally, positive correlations between all datasets occur in expected regions (as outlined in table B1) related to the specific index."

Line 416.  Changing "observed" to "observe" will made this statement present tense, like others in the section.

We changed to "observe" in the updated manuscript.

Line 425.  Delete "partly" or else explain the differences more fully.

We deleted "partly" in the updated manuscript.

Line 430. State the reference data sets used to generate the metrics since they are fundamental to the results.

We propose to change the sentence from line 427-429 into:

"Also, new versions of HOAPS (here: HOAPS V4, in phase 1: HOAPS V3.2) and ERA (here: ERA5, in phase 1: ERA-Interim) served as references for breakpoint detection and different SST (here: SST from ESA CCI, corrected v2, in phase 1: OI SST from NOAA, v2) and T_2m data records (not at all in phase 1) were employed for regression analysis."

Line 440.  Are 'expected ranges' determined by the Clausius-Clapeyron relationship?  In any case, the reason should be given.

True - a proper reasoning was missing. We added the following to section 3.2, line 225:

"Saturation vapour pressure is a function of air temperature and change in air temperature (see e.g. Hyland and Wexler, 1983). Saturation vapour pressure can be transferred into a change in mixing ratio assuming constant relative humidity and pressure. For a temperature change of 1 K the expected change in mixing ratio is between 6% at 300 K and 7.5% at 275 K."

and to change the sentence in section 5, starting line 444 into:

"It is recalled that the relationship between TCWV and surface temperature is affected by advection, precipitation, and other small-scale and regional events, which impact equilibrium between surface and atmosphere. Also, surface temperature and TCWV instead of near-surface air temperature and mixing ratio are considered here (e.g., Mieruch et al., 2014). Violations of these assumptions can give reason to larger than expected regression values (Trenberth, 2005}."

Line 444. Shi (2018) should be Shi et al.

Indeed, this has been fixed.

Line 445. Not clear what is meant by 'that time.'

It is indeed not very clear. "With that" refers to "event". We propose to delete this as it should be clear that a lag that depends on an event naturally depends on time as well.

Line 465. See earlier comments about diurnal sampling with sun-synchronous satellites.

We enhanced the information and the discussion (see above).

Thank you for the all the careful work!

Eric Fetzer

Thanks a lot, also for the detailed review.

---

## Author Comment (AC2)

**Authors' Response to Reviews of**
**Evaluation of Total Column Water Vapour Products from Satellite Observations and Reanalyses within the GEWEX Water Vapor Assessment**

Tim Trent, Marc Schroeder, Shu-Peng Ho, Steffen Beirle, Ralf Bennartz, Eva Borbas, Christian Borger, Helene Brogniez, Xavier Calbet, Elisa Castelli, Gilbert P. Compo, Wesley Ebisuzaki, Ulrike Falk, Frank Fell, John Forsythe, Hans Hersbach, Misako Kachi, Shinya Kobayashi, Robert E. Kursinsk, Diego Loyola, Zhengzao Luo, Johannes K. Nielsen, Enzo Papandrea, Laurence Picon, Rene Preusker, Anthony Reale, Lei Shi, Laura Slivinski, Joao Teixeira, Tom Vonder Haar, and Thomas Wagner

EGUsphere, https://doi.org/10.5194/egusphere-2023-28087, 2023

Please find below our responses to Reviewer 2, We include the original comments in black, our comments in green, and any alterations to text in blue.

I thoroughly enjoyed reviewing this manuscript. The authors work with an interesting dataset of global total column water vapour measurements from satellite observations and reanalysis within the GEWEX water vapour assessment. The evaluation was conducted through multiple approaches, some discrepancies and biases are observed, particularly in regions with complex topography or under certain meteorological conditions.

I believe that the manuscript addresses a relevant topic and includes a timely discussion. This is a well-written manuscript that only needs to undergo a few minor changes in addition to the other reviewers' comments:

Thanks for this feedback.

1. L193: please justify the reason for conducting this evaluation based on the monthly mean at 2° x 2°, since the coarse resolution may overlook some discrepancies among the datasets.

   This is the common resolution from version 1 of the archive, which we retain for consistency across the archive versions.

2. L296: please briefly explain the reasons that there is a "significant disagreement" between the datasets for dry atmosphere

   The observed disagreement will come from two main sources; i) either the satellite records have poor sensitivity to low column amounts of water vapour, or ii) reanalyses do not have sufficient in situ measurements to constrain them (very common in these regions). We have updated the final part of the paragraph to reflect this:

"Therefore, this highlights a significant disagreement between the archive records for dry atmospheres, especially at high latitudes. This disagreement can be driven by either low sensitivity in observational satellite records or a lack of in situ measurements to constrain reanalyses"

3. L339: missing cross-reference for the figure.

A cross-reference to Figure 2 is included now.

Thank you for taking the time to review our manuscript and your positive comments.

---

## Author Comment (AC3)

**Authors' Response to Reviews of**
**Evaluation of Total Column Water Vapour Products from Satellite Observations and Reanalyses within the GEWEX Water Vapor Assessment**

Tim Trent, Marc Schroeder, Shu-Peng Ho, Steffen Beirle, Ralf Bennartz, Eva Borbas, Christian Borger, Helene Brogniez, Xavier Calbet, Elisa Castelli, Gilbert P. Compo, Wesley Ebisuzaki, Ulrike Falk, Frank Fell, John Forsythe, Hans Hersbach, Misako Kachi, Shinya Kobayashi, Robert E. Kursinsk, Diego Loyola, Zhengzao Luo, Johannes K. Nielsen, Enzo Papandrea, Laurence Picon, Rene Preusker, Anthony Reale, Lei Shi, Laura Slivinski, Joao Teixeira, Tom Vonder Haar, and Thomas Wagner

EGUsphere, https://doi.org/10.5194/egusphere-2023-28087, 2023

Please find below our responses to the community comments, We include the original comments in black, our comments in green, and any alterations to text in blue.

This is an important and well constructed study that should in my opinion be published. It's impact could be improved with enhanced motivation to bring out the importance and some more direction to the community in terms of more/less reliable products and strengths/weaknesses or limitations and recommending more or less suitable applications. I just have minor comments listed below.

1) L14 ice free regions change with the season and year - will this alias (slightly) into the variability? TWCV --> TCWV

Our approach is to use the sea-ice mask to remove any grid cells where we have found sea ice throughout the 1988-2014 period, so it is invariant over time in that sense. We also do not use data beyond ±60 degrees. Therefore, the analysis does not consider areas where we would observe this effect. The sea-ice mask then modifies this latitude band range consistently across all records.

2) L15 Why are % changes considered in the fit to temperature but kg/m^2 in the trends? Although ranges are shown, it would be useful to the community to have some expert judgement, such as removing obviously spurious datasets (what is the expected physical range?)

Results from sensitivity to temperature are shown in %/K as the expectation are changes between 6-7.5 %/K as outlined in the new section 3.3.

As in the first phase of G-VAP we only analyse trends in absolute units and per decade. The main motivation is not climate change analysis but trend estimation as a tool to intercompare and characterise data records.

If trends in %/decade are shown a comparison to results from sensitivity to temperature is still not straightforward because results from the sensitivity analysis are computed via regression and not via trend ratios (see also discussion of point 19).

3) L23 seems to be missing text. Also Forster et al. (2021) (IPCC Chapter 7) deals more with radiative effects of water vapour. Some mention of recent updates in the field of water vapour and climate would strengthen the context and motivation of the study e.g. Colman & Soden (2021) RevModPhys doi:10.1103/RevModPhys.93.045002; Allan et al. (2022) JGR doi:10.1029/2022JD036728; Ding et al. (2022) LNEE doi:10.1007/978-981-19-2588-7_27; Douville et al. (2022) Comm. Earth Env. doi:10.1038/s43247-022-00561-z; Wu et al. (2024) GRL doi:10.1029/2023GL107909; Wan et al. (2024) HESS doi:10.5194/hess-2023-301 which build on previous assessments e.g. Trenberth et al. (2005) Clim. Dyn. doi:10.1007/s00382-005-0017-4

We thank you the comment and point us to these references. We have incioprated some of these into updates withihn the introduction.

4) L25 water vapour feedback magnitude should be updated to the latest IPCC report chapter (Forster et al. 2021). It should also note that mid and upper tropospheric water vapour is more important to the feedback strength than lower tropospheric changes that column integrated water vapour is more closely related to. There is however an important link between column integrated water vapour and precipitation as well as downward longwave radiation and atmospheric absorption of sunlight, both of which also impact the energy-water Tim, thank him say we have added some of these to text to enhance the manuscriptcycle coupling (e.g. Douville et al. 2021 IPCC; Fowler et al. 2020 Nature Rev Earth Sci. doi:10.1038/s43017-020-00128-6).

We thank you for highlighting the update in the magnitude of water vapour feedback and have updated the text inline with Forster et al. 2021. We also thank you for the additional comments and references, which we have used to update the introduction

5) L26 water vapour feedback magnitude is not "compared" to greenhouse gas forcing

This statement has been modified as part of updates made to the introduction.

6) L60 The 'long period' was also presumably chosen to commence at the start of the SSM/I record and for consistency with previous analyses e.g. Allan et al. (2020) NYAS doi:10.1111/nyas.14337

The start was chosen as a compromise between number of data records and maximised length. Thus, we decided already in the first phase of G-VAP to start the "long-term period" in 1988.

7) L75 - are the AIRS + AMSU v6 Obs4MIP data set (Tian & Hearty, 2020 Earth & Space Science, doi:10.1029/2020EA001438) version also evaluated? These were developed to remove systematic biases and allow better comparison with CMIP simulations.

No we only used the science products for consistency with other data records used in the study.

8) L102 do all products vertically integrate to the top of atmosphere or are some cut off at a certain level?

All are vertically integrated to TOA values

9) L105 I didn't understand "lower, respectively higher spatial resolution"

In order to avoid confusion we included the actual resolutions and removed "lower" and "higher".

10) L122 it would be useful to mention limitations of the datasets. For example ERA5 and other reanalyses are subject to a changing observing system that can introduce spurious changes, though water vapour now seems quite robust in ERA5 after the mid-1990s (e.g. Allan et al. 2022). The 20CR only assimilates SST and surface pressure so water vapour is determined by the model based on these contraints and so is for all intents and purposes an atmospheric model "amip" type simulation nudged towards realistic atmospheric circulation. For satellite datasets, degradation in sensors, orbital drifts and intercallibration present a challenge

We do not detail performance limitations for every record within the assessment, as these are usually documented elsewhere. Instead, we test the performance of data records and where issues are identified, investigate them further. We have added additional text to further explain the process we follow on the assessment:

"It should be noted that all water vapour records will have limitations based on their underlying assumptions or operational frameworks. For example, satellite sensors can experience degradation (often corrected through recalibration efforts e.g. Tabata et al. (2019)), reducing the sensitivity of an instrument, while reanalysis records can experience introduce shifts in the time series due to changes in observing systems assimilated (Schröder et al., 2017; Allan et al., 2022). Individual data record performance assessments are usually detailed in publications or via technical documents such as the Product User Guide (PUG) or Validation Report (VR) and are not provided here. Through the assessment, we can highlight performance issues (e.g. breakpoints) and attempt to map them to know issues. Where we cannot identify the cause, our results can be used by the data record teams in future product updates."

11) L204 - is a consistent land/sea/ice mask (e.g. Fig. 2) applied to all datasets (if not, this could introduce differences in variability). Is the mask a climatology as in Figure 2 (though ice varies seasonally and interannually) or does it vary from month to month?

See comments above

12) Figure 3 - is this a median across datasets? A fuller caption may help

This is the median across all datasets that span the common long-period, the caption has been modified to make this clearer to readers

13) Figure 4 - which of these correlations are significant or not?

Figure 4 has been updated to highlight which correlations are significant. This is presented a s heatmap to compliment the exisiting cortrelation plot.

14) L293 - seasonal range usually means range over the year but I think intra-seasonal range is meant?

Thank you for spotting this; the text has been updated accordingly.

15) L305 - large (e.g. 2-sigma deviations) could usefully be reported to suggest outliers

An addition of +/- 2 sigma lines have been added to figure 6

16) L312 ERA5 does not seem significantly wetter (e.g. probably depends on years chosen)

Yes, this could be the reason. However, we only analyse set time periods within the assessment and do not have scope to extend the analysis at this time. This definitely something we should consider in the future.

17) L317 IR estimates presumably sample clear-sky regions which are systematically drier than cloudy regions e.g. John et al. (2011) JGR doi:10.1029/2010JD015355 (presumably visible records are also susceptible). I think this is discussed later but could be flagged earlier.

This forms part of the clear sky bias discussion in section 5.

18) L339 missing reference

This has been rectified.

19) L343 Are these annual trends? Were trends in %/decade also computed? This could remove mean bias effects (e.g. wetter datasets may vary more in absolute terms but not percent) and it would be useful to quote % changes for consistency with other analysis (e.g. sensitivity to temperature) and the literature

As in the first phase of G-VAP we only analyse trends in absolute units and per decade. The main motivation is not climate change analysis but trend estimation as a tool to intercompare and characterise data records.

If trends in %/decade are shown a comparison to results from sensitivity to temperature is still not straightforward because results from the sensitivity analysis are computed via regression and not via trend ratios.

20) L347 is this the interannual regression or does it include the seasonal cycle (which is determined by very different processes)? Or is it the trend in TCWV divided by the trend in temperature? For example in Allan et al. (2022) the ERA5 global TCWV sensitivity to T2m is 5.76 +- 0.35%/K for 1988-2014 while the trend is 0.78+-0.08 %/decade which combined with a warming of 0.17 K/decade gives a lower sensitivity of 4.6 %/K. It was also noted that ERA5 decreases in TCWV over the ocean before the mid-1990s are at odds with the SSM/I record. Ocean and land estimates are also available in the paper.

For trend estimates we fit the annual cycle and ENSO strength simultaneously, while for the regression we remove the seasonal cycle from both the surface temperature and water vapour data. We understand that this may give different answers compared to other studies; however, for the assessment we do not claim that these are climate estimates, rather estimates of performance. This also keeps the analysis between different phases of the assessment consistent.

21) Figure 9 - if microwave values are masked does this mean there are variable numbers of datasets in each grid point? MIssing reference in caption.

This is indeed true. We did not include a corresponding statement in the caption as it should be obvious and because it does not impact the interpretation of the figure. The reference now points to Figure 2.

22) Figure 10/11 could be combined (and enlarged). It may also be useful to have a zoom in on the more homogeneous datasets since the outliers dominate somewhat

We may either include a single column plot as it is now or a double column plot. We may change to double column and ask the editor if this is acceptable. Else, we prefer to have separate figures to emphasise the focus of land versus ocean results.

23) L395 do any of the ERA5 breakpoints coincide with the early 1990s low latitude ocean trends identified in Allan et al. (2020, 2022) and Hersbach et al. (2020) that were also in previous versions of this dataset and linked with changes in the observing system? These seem linked with decreases in surface relative humidity and 850 hPa specific humidity over the ocean in the late 1980s-early 1990s.

Within the scope of our study, we are not able to determine regional trends. Therefore, we only focus on analysing global trends. Previously, these were done for global ice-free ocean only, but this time, we have also included land estimates. We think this would be an interesting avenue for future research and will consider

24) L402 - can changes in the mix of SSM/I satellites introduce spurious variability since they are observing at different overpass times. Some studies use particular SSM/I satellites with more stable or consistent overpass times to avoid this (e.g. Allan et al. 2022).

In general, a mix of satellites may introduce spurious variability even when carefully intercalibrated and optimally merged. One of the main aspects of the results presented in this paper is actually the impact of a changing observing on homogeneity. In line 402 we conclude that the observed break point does not coincide with a change of the observing system. However, we did not analyse if changes of observing system impact variability. We propose to add in line 402: "It can be a topic of future G-VAP efforts to analyse this feature further, e.g., by comparing the full SSM/I and SSMIS climatology to a climatology of near constant equator crossing times (similar as in Allan et al., 2022)".

25) Fig. 12 - the stippling seems to show discontinuity south of Alaska and south of India?

On closer examination we do not see this effect, we did notice issues on some draft printouts but not in the electronic pdf. Thus, we cannot clarify this issue.

26) L435 but only some of the breakpoints are matched to phyiscal causes?

Within the scope of this study, we are only able to identify most of the causes of the observed breakpoints. Where we were unable to identify a cause we mark these as 'Unclear'. We still include the information to 1) inform potential users, and 2) so data product teams can investigate these in greater depth during product updates. We update the sentence to be clearer:

"Most data records are affected by breakpoints, where some of the physical causes can be identified."

27) L460 although clear-sky sampling introduces dry biases, it only affects moisture variability if the clear-sky regions vary in a different way to the cloudy regions (e.g. Allan et al. 2003 QJRMetS doi:10.1256/qj.02.217)

Thanks for this comment. We did not explicitly analyse this aspect, except that we looked at spatial variability of the bias and at its annual dependencies. In both cases, we observed changes from month-to-month changes and regions to region. This variability is usually driven by changes in cloudy patterns.

28) Conclusion - the impact of this considerable work could be enhanced with some recommendations to the community with regard to better and worse datasets for particular applications (e.g. climatological, regional variability, interannual variability and long-term trends).

This part is outside the scope of the exercise. We indirectly do this by showing the results, this is intentionally left to the reader and depends on the type of analysis people do. We provide the information for users to make their decision

Richard Allan

We thank you the time taken to review the manuscript and many helpful comments and suggestions.